# Learning Temporally Causal Latent Processes from General Temporal Data

**Weiran Yao**[†*]  **Yuewen Sun**[‡*]  **Alex Ho**[◇]  **Changyin Sun**[‡]  **Kun Zhang**[†•]

[†]Carnegie Mellon University, Pittsburgh PA, USA

[‡]Southeast University, Nanjing, China

[◇]Rice University, Houston TX, USA

[•]Mohamed bin Zayed University of Artificial Intelligence, Abu Dhabi, United Arab Emirates

## Abstract

Our goal is to recover time-delayed latent causal variables and identify their relations from measured temporal data. Estimating causally-related latent variables from observations is particularly challenging as the latent variables are not uniquely recoverable in the most general case. In this work, we consider both a nonparametric, nonstationary setting and a parametric setting for the latent processes and propose two provable conditions under which temporally causal latent processes can be identified from their nonlinear mixtures. We propose LEAP, a theoretically-grounded framework that extends Variational AutoEncoders (VAEs) by enforcing our conditions through proper constraints in causal process prior. Experimental results on various datasets demonstrate that temporally causal latent processes are reliably identified from observed variables under different dependency structures and that our approach considerably outperforms baselines that do not properly leverage history or nonstationarity information. This demonstrates that using temporal information to learn latent processes from their invertible nonlinear mixtures in an unsupervised manner, for which we believe our work is one of the first, seems promising even without sparsity or minimality assumptions.

## 1 Introduction and Related Work

Causal discovery seeks to identify the underlying structure of the data generation process by exploiting an appropriate class of assumptions (Spirtes et al., 1993; Pearl, 2000). Despite its success in certain domains, most existing work either focuses on estimating the causal relations between observed variables (Spirtes & Glymour, 1991; Chickering, 2002; Shimizu et al., 2006), or starts from the premise that causal variables are given beforehand (Spirtes et al., 2013). Real-world observations (e.g., image pixels, sensor measurements, etc.), however, are not structured into causal variables to begin with. Estimating latent causal variable graphs from observations is particularly challenging as the latent variables, even with independent factors of variation (Locatello et al., 2019), are not identifiable or "uniquely" recoverable in the most general case (Hyvärinen & Pajunen, 1999). There exist several pieces of work aiming to uncover causally related latent variables. For instance, by exploiting the vanishing Tetrad conditions (Spearman, 1928) one is able to identify latent variables in linear-Gaussian models (Silva et al., 2006), and the so-called Generalized Independent Noise (GIN) condition was proposed to estimate linear, non-Gaussian latent variable causal graph (Xie et al., 2020), with follow-up studies such as (Adams et al., 2021). However, these approaches are constrained within linear relations, need certain types of sparsity or minimality assumptions, and require a relatively large number of measured variables as children of the latent variables. The work of (Bengio et al., 2019; Ke et al., 2019) used "quick adaptation" as the training criteria for learning latent structure but the identifiability results have not been theoretically established yet.

Recent advances in the theory of nonlinear Independent Component Analysis (ICA) have proven strong identifiability results (Hyvarinen & Morioka, 2016; 2017; Hyvarinen et al., 2019; Khemakhem et al., 2020; Sorrenson et al., 2020) by exploiting certain side information in addition to independence. By assuming that the generative latent factors $z_i$ are conditionally independent

---

\* Equal contribution. Code: https://github.com/weirayao/leap

given auxiliary variables $\mathbf{u}$ that may be time index, domain index, class label, etc. and augmenting observation data $\mathbf{x}$ with $\mathbf{u}$, deep generative models fit with such tuples $(\mathbf{x}, \mathbf{u})$ may be identifiable in function space; they can recover *independent factors* up to a certain transformation of the original latent variables under proper assumptions (note that we use "latent factor" and "latent processe" interchangeably). Although the temporal structure is widely used for nonlinear ICA, existing work that establishes identifiability results considers only independent sources, or further with linear transitions. However, these assumptions may severely distort the results if the real latent factors have causal relations in between, or if the relations are nonlinear. It is not clear yet how the temporal structure may help in learning *temporally causally-related latent factors*, together with their causal structure, from temporal observation data.

In this paper, we focus on the scenario where the observed temporal variables $\mathbf{x}_t$ do not have direct causal edges but are generated by latent processes or confounders $\mathbf{z}_t$ that have time-delayed causal relations in between. That is, the observed data $\mathbf{x}_t$ are unknown nonlinear (but invertible) mixtures of the underlying sources: $\mathbf{x}_t = g(\mathbf{z}_t)$. Our first goal is hence to understand under what conditions the latent temporally causal processes can be identified. Inspired by real situations, we consider both a nonparametric, nonstationary setting and a parametric setting for the latent processes. In the nonparametric setting, the generating process of each latent causal factor $z_{it}$ is characterized by nonparametric assignment $z_{it} = f_i(\mathbf{Pa}(z_{it}), \epsilon_{it})$, in which the parents of $z_{it}$ (i.e., the set of latent factors that directly cause $z_{it}$) together with noise term $\epsilon_{it} \sim p_\epsilon$ (where $p_\epsilon$ denotes the distribution of $\epsilon_{it}$) generate $z_{it}$ via unknown nonparametric function $f_i$ with some time delay. In the parametric setting, the time-delayed causal influences among latent factors follow a linear form. In both settings, we establish the identifiability of the latent factors and their causal influences, rendering them recoverable from the observed data.

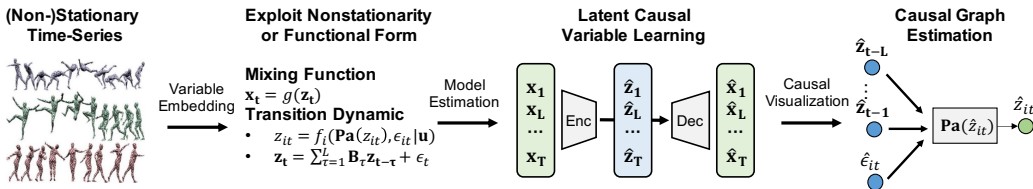

Figure 1: **Our approach:** we leverage nonstationarity in process noise or functional and distributional forms of temporal statistics to identify temporally causal latent processes from observation.

Our second goal is then to develop a theoretically-grounded training framework that enforces the assumed conditions through proper constraints. To this end, we propose Latent tEmporally cAusal Processes estimation (**LEAP**), a novel architecture that extends VAEs with a learned causal process prior network that enforces the Independent Noise (IN) condition and models possible nonstationarity through flow-based estimators. We evaluate LEAP on a number of synthetic and real-world datasets, including video and motion capture data with the properties required by our conditions. Experimental results demonstrate that temporally causal latent processes are reliably identified from observed variables under different dependency structures, and that our approach considerably outperforms existing methods which do not leverage history or nonstationarity information.

The closest work to ours includes (Klindt et al., 2020; Hyvarinen & Morioka, 2017), which require the underlying sources to be mutually independent for identifiability. Our work extends the theories for the discovery of conditionally independent sources to the setting with temporally causally-related latent processes, by leveraging nonstationarity or functional and distributional constraint on temporal relations. To the best of our knowledge, this is one of the first works that successfully recover time-delayed latent processes from their nonlinear mixtures without using sparsity or minimality assumptions. The proposed framework may serve as an alternative tool for creating versatile models robust to domain shifts and may be extended for more general conditions with changing causal relations over time.

## 2 IDENTIFIABILITY THEORY

### 2.1 IDENTIFIABILITY PROPERTY

We summarize the recent literature related to our work from four perspectives and compare our proposed theory with them in Table 1. The detailed comparisons of the problem settings are given in Appendix A.4. Following prior work, we define identifiability in representation function space.

Table 1: Attributes of existing theories. A green check denotes that a method has an attribute, whereas a red cross denotes the opposite. [†] indicates an approach we implemented.

| Approach | Temporal Data | Causally-related Factors | Nonparametric Expression | Stationary Process |
|---|---|---|---|---|
| TCL (Hyvarinen & Morioka, 2016) | ✓ | ✗ | ✗ | ✗ |
| PCL (Hyvarinen & Morioka, 2017) | ✓ | ✗ | ✓ | ✓ |
| GCL (Hyvarinen et al., 2019) | ✓ | ✗ | ✓ | ✗ |
| iVAE (Khemakhem et al., 2020) | ✗ | ✗ | ✗ | ✗ |
| GIN (Sorrenson et al., 2020) | ✗ | ✗ | ✗ | ✗ |
| HM-NLICA (Hälvä & Hyvarinen, 2020) | ✓ | ✗ | ✓ | ✗ |
| SlowVAE (Klindt et al., 2020) | ✓ | ✗ | ✗ | ✓ |
| CausalVAE (Yang et al., 2021) | ✗ | ✓ | ✗ | ✗ |
| **LEAP (Theorem 1)** [†] | ✓ | ✓ | ✓ | ✗ |
| **LEAP (Theorem 2)** [†] | ✓ | ✓ | ✗ | ✓ |

**Definition 1 (Componentwise Identifiability)** *Let $\mathbf{x}_t$ be a sequence of observed variables generated by the true temporally causal latent processes specified by $(f_i, p_{\epsilon_i})$ and nonlinear mixing function $g$, given in the introduction. A learned generative model $(\hat{g}, \hat{f}_i, \hat{p}_{\epsilon_i})$ is observationally equivalent to $(g, f_i, p_{\epsilon_i})$ if the joint distribution $p_{\hat{g},\hat{f},\hat{p}_\epsilon}(\mathbf{x}_t)$ matches $p_{g,f,p_\epsilon}(\mathbf{x}_t)$ everywhere. We say latent causal processes are identifiable if observational equivalence can always lead to identifiability of the latent variables up to permutation $\pi$ and component-wise invertible transformation $T$:*

$$p_{\hat{g},\hat{f},\hat{p}_\epsilon}(\mathbf{x}_t) = p_{g,f,p_\epsilon}(\mathbf{x}_t) \Rightarrow \hat{g} = g \circ T \circ \pi. \tag{1}$$

Once the latent causal processes are identifiable up to componentwise transformations, latent causal relations are also identifiable because conditional independence relations fully characterize time-delayed causal relations in a time-delayed causally sufficient system. Note that invertible componentwise transformations on latent causal processes do not change their conditional independence relations.

## 2.2 Our Proposed Conditions

We consider two novel conditions that ensure the identifiability of temporally causal latent processes using (1) nonstationarity or (2) functional and distributional constraints on their temporal relations. The corresponding identifiability of the latent processes is established in the following two theorems, with proofs and discussions of the assumed conditions provided in Appendix A.

**Theorem 1 (Nonparametric Processes)** *Assume nonparametric processes in Eq. 2, where the transition functions $f_i$ are third-order differentiable functions and mixing function $g$ is injective and differentiable almost everywhere; let $\mathbf{Pa}(z_{it})$ denote the set of (time-delayed) parent nodes of $z_{it}$:*

$$\underbrace{\mathbf{x}_t = g(\mathbf{z}_t)}_{\text{Nonlinear mixing}}, \quad \underbrace{z_{it} = f_i\left(\{z_{j,t-\tau}|z_{j,t-\tau} \in \mathbf{Pa}(z_{it})\}, \epsilon_{it}\right)}_{\text{Nonparametric transition}} \text{ with } \underbrace{\epsilon_{it} \sim p_{\epsilon_i|\mathbf{u}}}_{\text{Nonstationary noise}}. \tag{2}$$

*Here we assume:*

1. *(Nonstationary Noise): Noise distribution $p_{\epsilon_i|\mathbf{u}}$ is modulated (in any way) by the observed categorical auxiliary variables $\mathbf{u}$, which denotes nonstationary regimes or domain index;*

2. *(Independent Noise): The noise terms $\epsilon_{it}$ are mutually independent (i.e., spatially and temporally independent) in each regime of $\mathbf{u}$ (note that this directly implies that $\epsilon_{it}$ are independent from $\mathbf{Pa}(z_{it})$ in each regime);*

3. *(Sufficient Variability): For any $\mathbf{z}_t \in \mathbb{R}^n$ there exist $2n + 1$ values for $\mathbf{u}$, i.e., $\mathbf{u}_j$ with $j = 0, 1, ..., 2n$, such that the $2n$ vectors $\mathbf{w}(\mathbf{z}_t, \mathbf{u}_{j+1}) - \mathbf{w}(\mathbf{z}_t, \mathbf{u}_j)$, with $j = 0, 1, ..., 2n$, are linearly independent with $\mathbf{w}(\mathbf{z}_t, \mathbf{u})$ defined below, where $q_i$ is the log density of the conditional distribution and $\mathbf{z}_{Hx} = \{\mathbf{z}_{t-\tau}\}$ denotes history information up to maximum time lag $L$:*

$$\mathbf{w}(\mathbf{z}_t, \mathbf{u}) \triangleq \left( \frac{\partial q_1(z_{1t}|\mathbf{z}_{Hx}, \mathbf{u})}{\partial z_{1t}}, \dots, \frac{\partial q_n(z_{nt}|\mathbf{z}_{Hx}, \mathbf{u})}{\partial z_{nt}}, \frac{\partial^2 q_1(z_{1t}|\mathbf{z}_{Hx}, \mathbf{u})}{\partial z_{1t}^2}, \dots, \frac{\partial^2 q_n(z_{nt}|\mathbf{z}_{Hx}, \mathbf{u})}{\partial z_{nt}^2} \right). \tag{3}$$

*Then the componentwise identifiability property of temporally causal latent processes is ensured.*

**Theorem 2 (Parametric Processes)** *Assume the vector autoregressive process in Eq. 4, where the state transition functions are linear and additive and mixing function $g$ is injective and differentiable almost everywhere. Let $\mathbf{B}_\tau \in \mathbb{R}^{n \times n}$ be the state transition matrix at lag $\tau$. The process noises $\epsilon_{it}$*

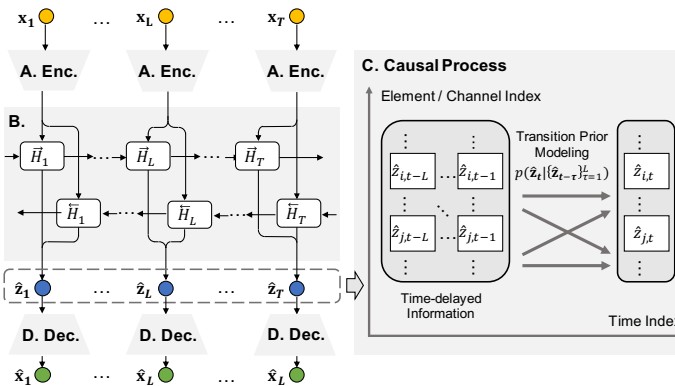

Figure 2: **LEAP**: Encoder (A) and Decoder (D) with MLP or CNN for specific data types; (B) Bidirectional inference network that approximates the posteriors of latent variables $\hat{\mathbf{z}}_{1:T}$, and (C) Causal process network that (1) models nonstationary latent causal processes $\hat{\mathbf{z}}_t$ with Independent Noise constraint (Thm 1) or (2) models the linear transition matrix with Laplacian constraints (Thm 2).

*are assumed to be stationary and both spatially and temporally independent:*

$$\underbrace{\mathbf{x}_t = g(\mathbf{z}_t)}_{\text{Nonlinear mixing}}, \quad \underbrace{\mathbf{z}_t = \sum_{\tau=1}^{L} \mathbf{B}_\tau \mathbf{z}_{t-\tau} + \epsilon_t}_{\text{Linear additive transition}} \ with \ \underbrace{\epsilon_{it} \sim p_{\epsilon_i}}_{\text{Independent noise}} . \tag{4}$$

*Here we assume:*

1. *(Generalized Laplacian Noise): Process noises $\epsilon_{it} \sim p_{\epsilon_i}$ are mutually independent and follow the generalized Laplacian distribution $p_{\epsilon_i} = \frac{\alpha_i \lambda_i}{2\Gamma(1/\alpha_i)} \exp\left(-\lambda_i |\epsilon_i|^{\alpha_i}\right)$ with $\alpha_i < 2$;*

2. *(Nonsingular State Transitions): For at least one $\tau$, the state transition matrix $\mathbf{B}_\tau$ is of full rank.*

*Then the componentwise identifiability property of temporally causal latent processes is ensured.*

## 3 LEAP: LATENT TEMPORALLY CAUSAL PROCESSES ESTIMATION

Given our identifiability results, we further propose a Latent tEmporally cAusal Processes (LEAP) estimation framework, which is built upon the framework of VAEs while enforcing the conditions in Section 2.2 as constraints for identification of the latent causal processes. The model architecture is shown in Fig. 2. Here $\mathbf{x}_{1:T}$ and $\hat{\mathbf{x}}_{1:T}$ are the observed and reconstructed time series, and $\overrightarrow{H}_t$ and $\overleftarrow{H}_t$ denote the forward and backward embeddings. Implementation details are in Appendix C.

### 3.1 CAUSAL PROCESS PRIOR NETWORK

We model latent causal processes in the learned prior network. To enforce the Independent Noise (IN) condition, latent transition priors $p(z_{it}|\mathbf{Pa}(z_{it}))$ are reparameterized into factorized noise distributions using the change of variable formula. To enforce the nonstationary noise condition, the noise distributions are learned by flow-based estimators, and independence constraints are enforced through the contrastive approach. Finally, for interpretation purposes of the causal relations, we use pruning techniques based on masked inputs and soft-thresholding.

#### 3.1.1 TRANSITION PRIOR MODELING

Nonparametric and parametric transition priors are modeled below, leading to two separate methods. The IN condition is used to reparameterize transition priors into factorized noise distributions.

**Nonparametric Transition** We propose a novel way to inject the IN condition into the transition prior. Let $\{r_i\}$ be a set of learned inverse causal transition functions that take the estimated latent causal variables and output the noise terms, i.e., $\hat{\epsilon}_{it} = r_i(\hat{z}_{it}, \{\hat{\mathbf{z}}_{t-\tau}\})$. We model each output component $\hat{\epsilon}_{it}$ with a separate Multi-Layer Perceptron (MLP) network, so we can easily disentangle the effects from inputs to outputs. Design transformation $\mathbf{A} \rightarrow \mathbf{B}$ with low-triangular Jacobian as follows:

$$\underbrace{\left[\hat{\mathbf{z}}_{t-L}, \ldots, \hat{\mathbf{z}}_{t-1}, \hat{\mathbf{z}}_t\right]^\top}_{\mathbf{A}} \text{ mapped to } \underbrace{\left[\hat{\mathbf{z}}_{t-L}, \ldots, \hat{\mathbf{z}}_{t-1}, \hat{\epsilon}_t\right]^\top}_{\mathbf{B}}, \ with \ \mathbf{J}_{\mathbf{A}\rightarrow\mathbf{B}} = \begin{pmatrix} \mathbb{I}_{nL} & 0 \\ * & \text{diag}\left(\frac{\partial r_i}{\partial \hat{z}_{it}}\right) \end{pmatrix}. \tag{5}$$

By applying the change of variables formula to the map from $\mathbf{A}$ to $\mathbf{B}$ and because of the IN condition in Assumption 2 of Theorem 1, one can obtain the joint distribution of the latent causal variables as:

$$\log p(\mathbf{A}|\mathbf{u}) = \log p(\mathbf{B}|\mathbf{u}) + \log\left(\left|\det\left(\mathbf{J}_{\mathbf{A}\to\mathbf{B}}\right)\right|\right) \tag{6}$$

$$= \underbrace{\log p\left(\hat{\mathbf{z}}_{t-L}, \ldots, \hat{\mathbf{z}}_{t-1}\right) + \sum_{i=1}^{n} \log p(\hat{\epsilon}_i|\mathbf{u})}_{\text{Because of the IN condition (See Assumption 2 in Thm 1)}} + \log\left(\left|\det\left(\mathbf{J}_{\mathbf{A}\to\mathbf{B}}\right)\right|\right). \tag{7}$$

The transition prior $p\left(\hat{\mathbf{z}}_t | \{\hat{\mathbf{z}}_{t-\tau}\}_{\tau=1}^{L}\right)$ can thus be evaluated using factorized noise distributions by cancelling out the marginals of time-delayed causal variables on both sides of Eq. 7. Given that this Jacobian is triangular, we can efficiently compute its determinant as $\prod_i \frac{\partial r_i}{\partial \hat{z}_{it}}$:

$$\log p\left(\hat{\mathbf{z}}_t | \{\hat{\mathbf{z}}_{t-\tau}\}_{\tau=1}^{L}, \mathbf{u}\right) = \sum_{i=1}^{n} \log p(\hat{\epsilon}_i|\mathbf{u}) + \sum_{i=1}^{n} \log \left|\frac{\partial r_i}{\partial \hat{z}_{it}}\right| \tag{8}$$

**Parametric Transition** A group of state transition matrices $\{\mathbf{r}_\tau\}$ is used to model the inverse transition functions: $\hat{\epsilon}_t = \hat{\mathbf{z}}_t - \sum_{\tau=1}^{L} \mathbf{r}_\tau \hat{\mathbf{z}}_{t-\tau}$. Because of additive noise, the transition priors can be directly written in terms of factorized noise distributions.

$$\log p\left(\hat{\mathbf{z}}_t | \{\hat{\mathbf{z}}_{t-\tau}\}_{\tau=1}^{L}\right) = \sum_{i=1}^{n} \log p(\hat{\epsilon}_t) \tag{9}$$

### 3.1.2 NONSTATIONARY NOISE ESTIMATION

A flow-based density estimator is used to fit the residuals (e.g., non-Gaussian noises) and score the likelihood in Eqs. 8 and 9. The independence of residuals is enforced inside the density estimator.

**Flow-based Noise Estimation** We apply the componentwise neural spline flow model (Dolatabadi et al., 2020) to fit the estimated noise terms. The distribution of each noise component $\epsilon_{it} \sim p_{\epsilon_i|\mathbf{u}}$ is modeled separately by transforming standard normal noises through linear rational splines $s_{i,\mathbf{u}}$. To model nonstationarity, we keep a copy of spline flows $\{s_{i,\mathbf{u}}\}$ for each nonstationary regime $\mathbf{u}$ and trigger it when data falls into this category: $\underbrace{p(\hat{\epsilon}_i|\mathbf{u}) = p_{\mathcal{N}(0,1)}\left(s_{i,\mathbf{u}}^{-1}(\hat{\epsilon}_i)\right)\left|\frac{ds_{i,\mathbf{u}}^{-1}(\hat{\epsilon}_i)}{d\hat{\epsilon}_i}\right|}_{\text{Nonstationary noise condition}}$. Stationary sources are considered as special cases where the nonstationary regime $\mathbf{u} = 1$ for all data samples.

**Independence by Contrastive Learning** We further force the estimated noises $\{\hat{\epsilon}_{it}\}$ to be mutually independent (corresponding to the IN condition) across values of $(i, t)$. Similar to FactorVAE (Kim & Mnih, 2018), we train a discriminator $\mathcal{D}(\{\hat{\epsilon}_{it}\})$ together with the latent variable model, to distinguish positive samples which are the estimated noise terms, against negative samples $\{\hat{\epsilon}_{it}^{\text{perm}}\}$ which are random permutations of the noise terms across the batch for each noise dimension $(i, t)$. The Total Correlation (TC) of noise terms can be estimated by the density-ratio trick and is added to the Evidence Lower BOund (ELBO) objective function to enforce joint independence of noise terms: $\mathcal{L}_{TC} = \mathbb{E}_{\{\hat{\epsilon}_{it}\}\sim(q(\hat{\mathbf{z}}_t), r_i)} \log \frac{\mathcal{D}(\{\hat{\epsilon}_{it}\})}{1 - \mathcal{D}(\{\hat{\epsilon}_{it}\})}$.

### 3.1.3 STRUCTURE ESTIMATION

For visualization purposes, we apply sparsity-encouraging regularizations to the learned causal relations in latent processes. Specifically, we use a combination of masked inputs and pruning approaches. Note that our identifiability results do not rely on sparsity of causal relations in latent processes. It is used only for visualizing causal relations when the causal processes are nonlinear.

**Masked Input and Regularization** For latent processes of sparse causal relations, each MLP of the inverse transition function $r_i$ has a learned n-dimensional soft mask vector $\sigma(\gamma_i) \in [0, 1]$ with the $j$-th time-delayed inputs of the MLP being multiplied by $\sigma(\gamma_{ij})$. A fixed $L_1$ penalty is added to the mask during training: $\mathcal{L}_{\text{Mask}} = \|\sigma(\gamma_{ij})\|_1$. For linear transition, the penalty is added to the transition matrices since the weights directly indicate whether an edge exists.

**Pruning** For nonlinear relations, we use LassoNet (Lemhadri et al., 2021) as a post-processing step to remove weak edges. This approach prunes input nodes by jointly passing the residual layer and the first hidden layer through a hierarchical soft-thresholding optimizer. We fit the model on a

subset of the recovered latent causal variables. Though the true causal relations may not follow the causal additive assumption, the pruning step usually produces sparse causal relations.

## 3.2 INFERENCE NETWORK

A bidirectional Gated Recurrent Unit is used to infer latent variables. We approximate the posterior $q_\phi(\hat{\mathbf{z}}_{1:T}|\mathbf{x}_{1:T})$ with an isotropic Gaussian with mean and variance terms from the inference network. The KL divergence is $\mathcal{L}_{\text{KLD}} = D_{KL}(q_\phi(\hat{\mathbf{z}}_{1:T}|\mathbf{x}_{1:T})||p(\hat{\mathbf{z}}_{1:T}))$ and is estimated via sampling approach because prior distribution is not specified but rather learned by causal process network.

## 3.3 ENCODER AND DECODER

The reconstruction likelihood is $\mathcal{L}_{\text{Recon}} = p_{\text{reconstruct}}(\mathbf{x}_t|\hat{\mathbf{z}}_t)$, where $p_{\text{reconstruct}}$ is the decoder distribution. For synthetic and point cloud data, we use MLP with LeakyReLU units as encoder and decoder and MSE loss is used for reconstruction. For video datasets with single objects (e.g., KiT-TiMask), vanilla CNNs and binary cross-entropy loss are used. For videos with multiple objects, we apply a disentangled design (Kulkarni et al., 2019) with two separate CNNs, one for extracting visual features and the other for locating object locations with spatial softmax units. The decoder retrieves object features using object locations and reconstructs the scene with MSE loss. The network architecture details are given in Appendix C.1.

## 3.4 OPTIMIZATION

We train the VAE and noise discriminator jointly. The VAE parameters are updated using the augmented ELBO objective $\mathcal{L}_{\text{ELBO}}$. The discriminator is trained to distinguish between residuals from $q(\{\hat{\epsilon}_{it}\})$ and $q(\{\hat{\epsilon}_{it}^{\text{perm}}\})$ with $\mathcal{L}_{\mathcal{D}}$, thus learning to approximate the density ratio for estimating $\mathcal{L}_{\text{TC}}$:

$$\mathcal{L}_{\text{ELBO}} = \frac{1}{N}\sum_{i\in N}\mathcal{L}_{\text{Recon}} - \beta\mathcal{L}_{\text{KLD}} - \gamma\mathcal{L}_{\text{Mask}} - \sigma\mathcal{L}_{\text{TC}}, \ \mathcal{L}_{\mathcal{D}} = \frac{1}{2N}\sum_{i\in N}\left[\log\mathcal{D}(\{\hat{\epsilon}_{it}\}) + \sum_{i\in N'}\log\left(1 - \mathcal{D}(\{\hat{\epsilon}_{it}^{\text{perm}}\})\right)\right].$$

The discussions of the hyperparameter selection and sensitivity analysis are in Appendix C.2.1.

## 4 EXPERIMENTS

We comparatively evaluate LEAP on a number of temporal datasets with the required assumptions satisfied or violated. We aim to answer the following questions:

1. Does LEAP reliably learn temporally-causal latent processes from scratch under the proposed conditions? What is the contribution of each module in the architecture?
2. Is history/nonstationary information necessary for the identifiability of latent causal variables?
3. How do common assumptions in nonlinear ICA (i.e., independent sources or linear relations assumptions) distort identifiability if there are time-delayed causal relations between the latent factors, or if the latent processes are nonlinearly related?
4. Does LEAP generalize when some critical assumptions in the proposed conditions are violated? For instance, how does it perform in the presence of instantaneous or changing causal influences?

**Evaluation Metrics** To measure the identifiability of latent causal variables, we compute *Mean Correlation Coefficient* (MCC) on the validation dataset, a standard metric in the ICA literature for continuous variables. MCC reaches 1 when latent variables are perfectly identifiable up to permutation and componentwise invertible transformation in the noiseless case (we use Pearson correlation and rank correlation for linearly and nonlinearly related latent processes, respectively). To evaluate the recovery performance on causal relations, we use different approaches for (1) linear and (2) nonlinear transitions: (1) the entries of estimated state transition matrices are compared with the true ones after permutation, signs, and scaling are adjusted, and (2) the estimated causal skeleton is compared with the true data structure, and *Structural Hamming Distance* (SHD) is computed.

**Baselines and Ablation** We experimented with three kinds of nonlinear ICA baselines: **(1)** BetaVAE (Higgins et al., 2016) and FactorVAE (Kim & Mnih, 2018) which ignore both history and nonstationarity information; **(2)** iVAE (Khemakhem et al., 2020) and TCL (Hyvarinen & Morioka, 2016) which exploit nonstationarity to establish identifiability, and **(3)** SlowVAE (Klindt et al., 2020) and PCL (Hyvarinen & Morioka, 2017) which exploit temporal constraints but assume independent sources. Model variants are built to disentangle the contributions of different modules. As in Table 2, we start with BetaVAE and add our proposed modules successively without any change on the training settings. Finally, **(4)**, we fit our LEAP that uses linear transitions (LEAP-VAR) with nonstationary data to show if linear relation assumptions distort identifiability.

## 4.1 SYNTHETIC EXPERIMENTS

We first design synthetic datasets with the properties required by NonParametric **(NP)** and parametric Vector AutoRegressive **(VAR)** conditions in Section 2.2. We set latent size $n = 8$. The lag number of the process is set to $L = 2$. The mixing function $g$ is a random three-layer MLP with LeakyReLU units. We give the data generation procedures for nonparametric conditions, parametric conditions, and five types of datasets that violate our assumptions separately in Appendix B.1.

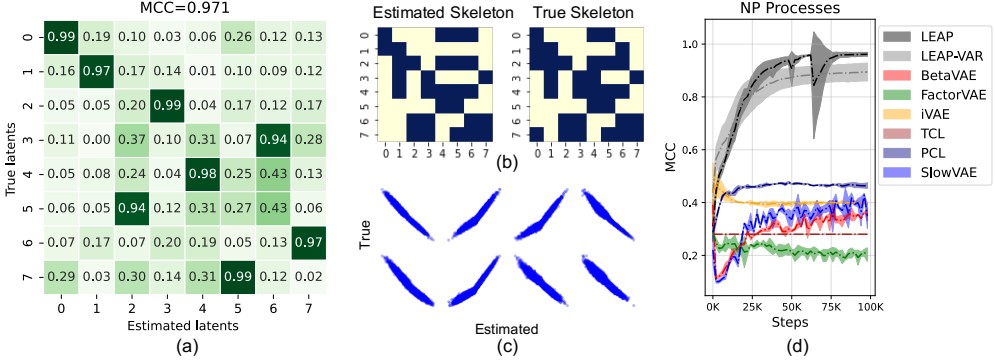

Figure 3: Results for synthetic nonparametric processes (NP) datasets: (a) MCC for causally-related factors; (b) recovered causal skeletons with (SHD=5); (c) scatterplots between estimated and true factors; and (d) MCC trajectories comparisons between LEAP and baselines.

**Main Results**   Fig. 3 gives the results on NP datasets. The latent processes are successfully recovered, as indicated by (a) high MCC for the casually-related factors and (b) the recovery of the causal relations (SHD=5). Panel (c) suggests that the latent causal variables are estimated up to permutation and componentwise invertible transformation. The comparisons with baselines are in (d). In general, the baselines that do not exploit history or nonstationarity cannot recover the latent processes. SlowVAE and PCL distort the results due to independent source assumptions. Interestingly, LEAP-VAR, which uses linear causal transitions, gains partial identifiability on NP datasets. This might promote the usage of linear components of transition signals to guide the learning of latent causal variables.

The results for VAR datasets are in Fig. 4. Similarly, the latent processes are recovered, as indicated by (a) high MCC and (b) recovery of state transition matrices. The latent causal variables are estimated up to permutation and scaling (c). The baselines without using history or assume independent sources again fail to recover the latent processes (d). We show the contributions of different

Table 2: Contribution of each module to MCC.

| Module | NP (Dense) | VAR |
|---|---|---|
| Baseline ($\beta$-VAE) | $0.446 \pm 0.004$ | $0.495 \pm 0.007$ |
| + Causal Process Prior | $0.721 \pm 0.121$ | $0.752 \pm 0.035$ |
| + Nonstationary Flow | $0.939 \pm 0.008$ | $0.935 \pm 0.014$ |
| + Noise Discriminator | $0.983 \pm 0.002$ | $0.978 \pm 0.004$ |

components of LEAP in Table 2. Causal process prior and nonstationary flow significantly improve identifiability. Noise discriminator further increases MCC and reduces variance. Note that we use a dense network without the input masks on the NP dataset for ablation studies, to validate that our proposed framework does not rely on sparse causal structure for latent causal discovery.

**Robustness**   We show the consequences of the violation of each of the assumptions on the synthetic datasets to mimic real situations. For VAR processes, we create datasets: (1) with causal relations changing over regime, and (2) with instantaneous causal relations, (3) with Gaussian noise, and (4) with low-rank state transition matrices. For NP processes, we violate (5) the sufficient variability by creating datasets with fewer than the required $2n + 1 = 17$ regimes. We fit LEAP on these datasets without any modification. Our framework can gain partial identifiability under (1,4) but violating (2) conditional independence and (3) non-Gaussianity distort the results obviously. Our approach, although designed to model nonstationarity by noise, can be extended to model changing causal relations. The partial identifiability of (4) is because the low-dimensional projections of the latent processes are recovered, as illustrated by a numerical example in Appendix A.2.4. For NP processes, nonstationarity is necessary for identifiability. Furthermore, the differences of the MCC trajectories under 15 and 20 regimes seem marginal, suggesting that our approach does not always require at least $2n + 1 = 17$ regimes to achieve full identifiability of the latent processes.

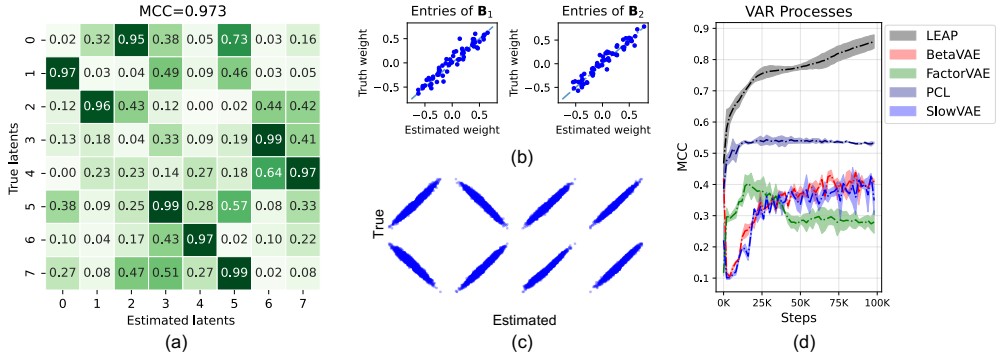

Figure 4: Results for synthetic parametric processes (VAR) datasets: (a) MCC for causally-related factors; (b) scatterplots of the entries of $\mathbf{B}_\tau$; (c) scatterplots between estimated and true factors; and (4) MCC trajectories comparisons between LEAP and baselines.

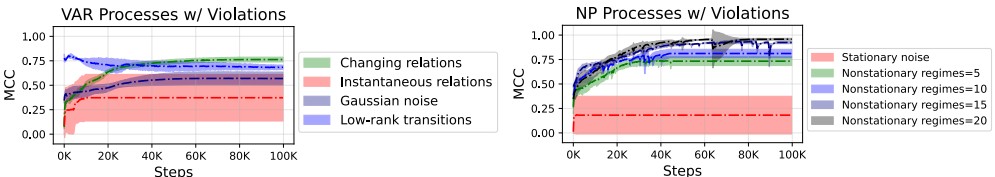

Figure 5: MCC trajectories of LEAP for temporal data with clear assumption violations.

## 4.2 REAL-WORLD APPLICATIONS: PERCEPTUAL CAUSALITY

Three public datasets including KiTTiMask (Klindt et al., 2020), Mass-Spring system (Li et al., 2020), and CMU MoCap database are used. The data descriptions are in Appendix B.2; depending on its property (e.g., whether it has multiple regimes), we apply the corresponding method. We first compare the MCC performances between our approach and the baselines on KiTTiMask and Mass-Spring system in Fig. 6. Our parametric method considerably outperforms the baselines that do not use history information. Because the true latent variables for CMU-MoCap are unknown, we visualize the latent traversals and the recovered skeletons in Appendix D.1, qualitatively comparing our nonparametric method with the baselines in terms of how intuitively sensible the recovered processes and skeletons are.

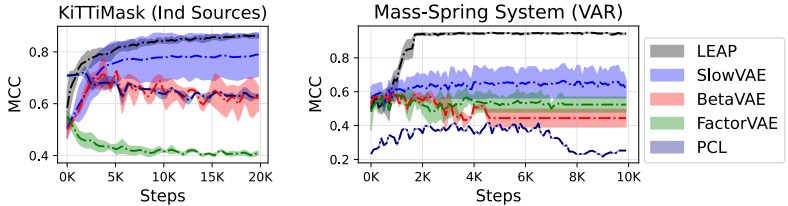

Figure 6: MCC trajectories comparisons on KiTTiMasks and Mass-Spring system.

**Parametric Transition – KiTTiMask** LEAP with VAR transitions is used. We set latent size $n = 10$ and the lag number $L = 1$. The gap between our result and that by SlowVAE is relatively small, as seen in Fig. 6; this is because the latent processes on this dataset seem rather independent (according to the transition matrix learned by LEAP, given in Fig. 7(c)) and when the latent processes are independent, our VAR method reduces to SlowVAE, as its special case. As shown in Fig. 7, the latent causal processes are recovered, as seen from (a) high MCC for independent sources; (b) latent factors estimated up to componentwise transformation; (c) the estimated state transition matrix, which is almost diagonal (independent sources); and (d) latent traversals confirming that the three latent causal variables correspond to the vertical, horizontal, and scale of pedestrian masks.

**Parametric Transition – Mass-Spring System** Mass-Spring system is a linear dynamical system with ball locations $(x_t^i, y_t^i)$ as state variables and lag number $L = 2$. LEAP with VAR transitions is used. In Fig. 8, the time-delayed cross causal relations are recovered as: (a) causal variables keypoint $(x_t^i, y_t^i)$ are successfully estimated; (b) the spring connections between balls are recovered (SHD=0). Visualizations of recovery of latent variables and skeletons are showcased in Appendix D.2.

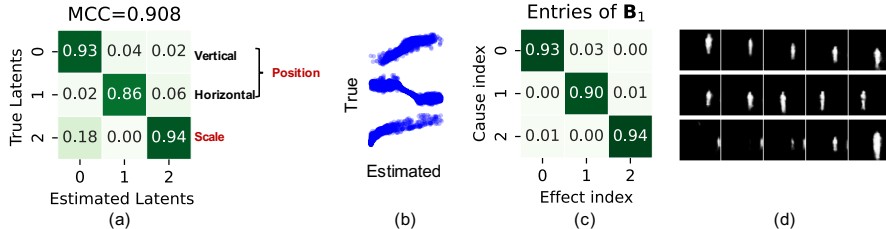

Figure 7: KiTTiMask dataset results: (a) MCC for independent sources; (b) scatterplots between estimated and true factors; (c) entries of $\mathbf{B}_1$; and (d) latent traversal on a fixed video frame.

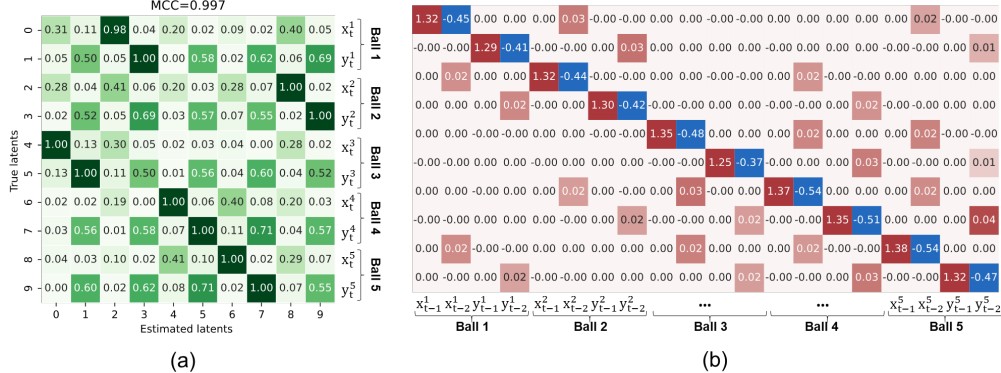

Figure 8: Mass-Spring system results: (a) MCC for causally-related sources; (b) entries of $\mathbf{B}_{1,2}$.

**Nonparametric Transition – CMU-MoCap**  We fit LEAP with nonparametric transitions on 12 trials of motion capture data for subject 7 with 62 observed variables of skeleton-based measurements at each time step. The 12 trials contain walk cycles with slightly different dynamics (e.g., walk, slow walk, brisk walk). We set latent size $n = 8$ and lag number $L = 2$. The differences between trials are modeled by nonstationary noise with one regime for each trial. The results are in Fig. 9. Three latent variables (which seem to be pitch, yaw, roll rotations, respectively) are found to explain most of the variances of human walk cycles (Panel c). The learned latent coordinates show smooth cyclic patterns with slight differences among trials (Panel a). Finally, we find that pitch (e.g., limb movement) and roll (e.g., shoulder movement) of human walking are coupled while yaw has independent dynamics (Panel b).

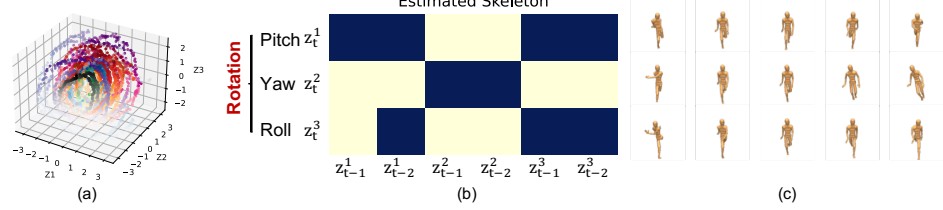

Figure 9: MoCap dataset results: (a) latent coordinates dynamics for 12 trials; (b) estimated skeleton; and (c) latent traversal by rendering the reconstructed point clouds into the video frame.

## 5   CONCLUSION AND FUTURE WORK

In this work, we proposed two provable conditions under which temporally causal latent processes can be identified from their observed nonlinear mixtures. The theories have been validated on a number of datasets with the properties required by the conditions. The main limitations of this work lie in our two major assumptions: (1) there is no instantaneous causal influence between latent causal processes, and (2) causal influences do not change across regimes. Both of them may not be true for some specific types of temporal data. The existence of instantaneous relations distorts identifiability results, but the amount of these relations can be controlled by time resolution. While we do not establish theories under changing causal relations, we have demonstrated through experiments the possibilities of generalizing our identifiability results to changing dynamics. Extending our identifiability theories and framework to accommodate such properties is our future directions.

## 6 Reproducibility Statement

Our code for the proposed framework and experiments can be found at https://github.com/weirayao/leap. For theoretical results, the assumptions and complete proof of the claims are in Appendix A. For synthetic experiments, the data generation process is described in Appendix B.1. The implementation details of our framework are given in Appendix C.

## Acknowledgement

KZ would like to acknowledge the support by the National Institutes of Health (NIH) under Contract R01HL159805, by the NSF-Convergence Accelerator Track-D award #2134901, and by the United States Air Force under Contract No. FA8650-17-C7715. YS and CS would like to acknowledge the support by the National Key R&D Program of China (No. 2018AAA0101400), National Natural Science Foundation of China (No. 61921004), and the Natural Science Foundation of Jiangsu Province of China (No. BK20202006).

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

# Supplement to "Learning Temporally Causal Latent Processes from General Temporal Data"

The supplementary materials are divided into five main sections. In Appendix A, we provide the explanations of each assumption and give the proof of the identifiability theory. We also give side-by-side comparisons, by providing the mathematical formulations of the closest works and making comparisons in terms of problem setups and critical assumptions. Finally, how the theory is connected to the training framework is discussed. In Appendix B, we provide the details of the synthetic and real-world datasets and explain the evaluation metrics. In Appendix C, we describe our network architecture, hyperparameters setting, and training details. The additional experiment results are given in Appendix D. The related work is summarized in Appendix E.

# A IDENTIFIABILITY THEORY

## A.1 NOTATION AND TERMINOLOGY

We summarize the notations used throughout the paper in Table A.1.

Table A.1: List of notations.

| | |
|---|---|
| **Index** | |
| $t$ | Time index |
| $i, j$ | Variable element (channel) index |
| $\tau$ | Time lag index |
| perm | Random permutated variable index across the data batch |
| | |
| **Variable** | |
| $\mathbf{x}_t$ | Observation data |
| $\hat{\mathbf{x}}_t$ | Reconstructed observation |
| $\mathbf{u}$ | Auxiliary nonstationary regime variable |
| $\mathbf{z}_t$ | Underlying sources |
| $\mathbf{z}_{\mathrm{Hx}}$ | Time-delayed latent causal variables |
| $\mathbf{Pa}(z_{it})$ | Set of direct cause nodes/parents of node $z_{it}$ |
| $\mathbf{e}_t$ | Measurement error |
| $\overrightarrow{H}, \overleftarrow{H}$ | Forward or backward embeddings in bidirectional RNN |
| $\sigma(\gamma)$ | Soft mask vector |
| $\mathbf{w}$ | Modulation parameter vector |
| $\mathbf{B}$ | State transition matrix |
| $\epsilon_{it}$ | Process noise term |
| $\hat{\epsilon}_{it}$ | Estimated process noise term |
| $\hat{z}_{it}$ | Estimated sources |
| $z_{it}$ | True underlying sources |
| | |
| **Function and Hyperparameter** | |
| $p$ | Distribution function (e.g., $p_{\epsilon_{it}}$ is the distribution of $\epsilon_{it}$.) |
| $g$ | Arbitrary nonlinear and injective mixing function |
| $f_i$ | Nonlinear transition function for $z_{it}$ |
| $h$ | Indeterminacy mappings between $\mathbf{z}_t$ and $\hat{\mathbf{z}}_t$ |
| $r_i$ | Learned inverse transition function for residual $\hat{\epsilon}_i$ |
| $s_{i,\mathbf{u}}$ | Spline flow function for residual $\hat{\epsilon}_i$ in regime $\mathbf{u}$ |
| $\beta, \gamma, \sigma$ | Weights in the augmented ELBO objective |
| $n$ | Latent size |
| $L$ | Maximum time lag |
| $T$ | Total length of time series |
| $\pi$ | Permutation operation |
| $T$ | Component-wise invertible nonlinearities |
| $q$ | Log density function |
| $\lambda, \alpha$ | Parameters of Laplacian distribution |

## A.2 DISCUSSION OF OUR ASSUMED CONDITIONS

We first explain and justify each critical assumption in the proposed conditions. We then discuss how restrictive or mild the conditions are in real applications.

### A.2.1 INDEPENDENT NOISE (IN) CONDITION

The IN condition was introduced in the Structural Equation Model (SEM), which represents effect $Y$ as a function of direct causes $X$ and noise $E$:

$$Y = f(X, E) \quad with \quad \underbrace{X \perp\!\!\!\perp E}_{\text{IN condition}} . \tag{10}$$

If $X$ and $Y$ do not have a common cause, as seen from the causal sufficiency assumption of structural equation models in Chapter 1.4.1 of Pearl's book (Pearl et al., 2000), the IN condition states that the unexplained noise variable $E$ is statistically independent of cause $X$. IN is a direct result of assuming causal sufficiency in SEM. The main idea for the proof is that if IN is violated, then by the common cause principle (Reichenbach, 1956), there exist hidden confounders that cause their dependence, thus violating the causal sufficiency assumption. Furthermore, for a causally sufficient system with acyclic causal relations, the noise terms in different variables are mutually independent. The main idea is that when the noise terms are dependent, it is customary to encode such dependencies by augmenting the graph with hidden confounder variables (Pearl et al., 2000), which means that the system is not causally sufficient.

In this paper, we assume the underlying latent processes form a casually-sufficient system without latent causal confounders. Then, the process noise terms $\epsilon_{it}$ are mutually independent, and moreover, the process noise terms $\epsilon_{it}$ are independent of direct cause/parent nodes $\mathbf{Pa}(z_{it})$ because of time information (the causal graph is acyclic because of the temporal precedence constraint).

**Applicability**    Loosely speaking, if there are no latent causal confounders in the (latent) causal processes and the sampling frequency is high enough to observe the underlying dynamics, then the IN condition assumed in this paper is satisfied in a causally-sufficient system and, moreover, there is no instantaneous causal influence (because of the high enough resolution). At the same time, we acknowledge that there exist situations where the resolution is low and there appears to be instantaneous dependence. However, there are several pieces of work dealing with causal discovery from measured time series in such situations; see. e.g., Granger (1987); Gong* et al. (2015); Danks & Plis (2013); Gong et al. (2017). In case there are instantaneous causal relations among latent causal processes, one would need additional sparsity or minimality conditions to recover the latent processes and their relations, as demonstrated in Silva et al. (2006); Adams et al. (2021). How to address the issue of instantaneous dependency or instantaneous causal relations in the latent processes will be one line of our future work.

### A.2.2   Nonstationary Noise and Sufficient Variability Condition

**Nonstationary Noise**    For nonparametric processes, temporal constraints are not sufficient for the identification of latent causal transition dynamics whose functional or distributional form is not constrained. Otherwise, there is no need for Theorem 2 to assume the generalized Laplacian noise and the full-rankness of state transitions at all. In this paper, an alternative way is to exploit the (temporal) nonstationarity of the data caused by changing noise distribution (hence called nonstationary noise condition). We assume the functions of the temporal causal influences denoted by $f_i$ remain the same across across the $|\mathbf{u}|$ regimes or domains of data we have observed, but the distributions $p_{\epsilon_i|\mathbf{u}}$ of noise terms that serve as arguments to the structural equation models, may change. One special case of this principle uses nonstationary variances, i.e., the noise variances change across nonstationary regimes. This kind of perturbation has been widely used in linear ICA (Matsuoka et al., 1995). Additionally, the nonstationary noise condition in this paper allows for any kinds of modulation of noise distribution by nonstationary regimes $\mathbf{u}$, such as changing distributional forms, scale, and location by $\mathbf{u}$, as long as the modulated sources satisfy sufficient variability condition described below.

**Sufficient Variability**    The sufficient variability condition was introduced in GCL (Hyvarinen et al., 2019) to extend the modulated exponential families (Hyvarinen & Morioka, 2016) to general modulated distributions. Essentially, the condition says that the nonstationary regimes $\mathbf{u}$ must have a sufficiently complex and diverse effect on the transition distributions. In other words, if the underlying distributions are composed of relatively many domains of data, the condition generally holds true. For instance, in the linear Auto-Regressive (AR) model with Gaussian innovations where only the noise variance changes, the condition reduces to the statement in (Matsuoka et al., 1995) that the variance of each noise term fluctuates somewhat independently of each other in different nonstationary regimes. Then the condition is easily attained if the variance vector of noise terms in any regime is not a linear combination of variance vectors of noise terms in other regimes.

We further illustrate the condition using the example of modulated conditional exponential families in (Hyvarinen et al., 2019). Let the log-pdf $q(\mathbf{z}_t|\{\mathbf{z}_{t-\tau}\}, \mathbf{u})$ be a conditional exponential family

distribution of order $k$ given nonstationary regime $\mathbf{u}$ and history $\mathbf{z}_{\text{Hx}} = \{\mathbf{z}_{t-\tau}\}$:

$$q(z_{it}|\mathbf{z}_{\text{Hx}}, \mathbf{u}) = q_i(z_{it}) + \sum_{j=1}^{k} q_{ij}(z_{it})\lambda_{ij}(\mathbf{z}_{\text{Hx}}, \mathbf{u}) - \log Z(\mathbf{z}_{\text{Hx}}, \mathbf{u}), \tag{11}$$

where $q_i$ is the base measure, $q_{ij}$ is the function of the sufficient statistic, $\lambda_{ij}$ is the natural parameter, and $\log Z$ is the log-partition. Loosely speaking, the sufficient variability holds if the modulation of by $\mathbf{u}$ on the conditional distribution $q(z_{it}|\mathbf{z}_{\text{Hx}}, \mathbf{u})$ is not too simple in the following sense:

1. Higher order of $k$ ($k > 1$) is required. If $k = 1$, the sufficient variability cannot hold;

2. The modulation impacts $\lambda_{ij}$ by $\mathbf{u}$ must be linearly independent across regimes $\mathbf{u}$. The sufficient statistics functions $q_{ij}$ cannot be all linear, i.e., we require higher-order statistics.

Further details of this example can be found in Appendix B of (Hyvarinen et al., 2019). In summary, we need the modulation by $\mathbf{u}$ to have diverse (i.e., distinct influences) and complex impacts on the underlying data generation process.

**Applicability** The nonstationarity of process noise seems to be prominent in many kinds of temporal data. For example, nonstationary variances are seen in EEG/MEG, natural video, and closely related to changes in volatility in financial time series (Hyvarinen & Morioka, 2016). As we assume the transition functions $f_i$ are fixed across regimes, the data that most likely satisfy the proposed condition is a collection of multiple trials/segments of data with slightly different temporal dynamics in between, where the differences can be well modeled by different noise distributions. For instance, in MEG data, temporal nonstationarity can be modeled by segmenting the measured data into different sessions (e.g., stimuli, rest, etc.) where the session index modulates the noise variance.

### A.2.3 GENERALIZED LAPLACIAN NOISE CONDITION

In the parametric (VAR) processes in Theorem 2, we exploit the non-Gaussianity of noise perturbations to achieve identifiability. Specifically, we constrain the process noise distribution to be within the generalized Laplacian distribution family in this paper. This L1-sparse temporal prior is motivated by the natural statistics of video data, where the uncertainty could have sharp impacts on some latent factors, but most other factors are not perturbed between two adjacent frames. This transition prior has strong connections with slow feature analysis (Sprekeler et al., 2014; Klindt et al., 2020) which measures slowness in terms of the L2 distance between temporally adjacent encodings as temporal constraints for nonlinear ICA. Note that although the Laplacian-like distributional form is pre-defined, the generalized Laplacian distrbution can still be used to fit a broad family of perturbations with different shapes by changing $\alpha$ and $\lambda$ of the distribution.

**Applicability** L1-sparse transition priors are widely used to model video datasets and natural scene measurements. This condition is applicable to video datasets where the external factors have sharp effects on some but not all latent factors in two adjacent frames.

### A.2.4 NONSINGULAR STATE TRANSITIONS CONDITION

Nonsingularity is a standard assumption made in the previous studies (Zhang & Hyvärinen, 2011) to achieve identifiability of linear state-space models. We give a two-dimensional low-rank vector autoregressive (VAR) process example below to illustrate the concept. Define a low-rank VAR process with time lag $L = 1$ below:

$$\mathbf{z}_t = \mathbf{B}\mathbf{z}_{t-1} + \epsilon_t \quad with \quad \mathbf{B} = \begin{bmatrix} a & b \\ \alpha \times a & \alpha \times b \end{bmatrix} = \begin{bmatrix} 1 \\ \alpha \end{bmatrix} \times [a \quad b], \tag{12}$$

where $\mathbf{z}_t = [z_{1t}, z_{2t}]^\top$ and $\epsilon_t = [\epsilon_{1t}, \epsilon_{2t}]^\top$. Multiplying both sides of the VAR process with a row vector $[a, b]$, we have:

$$\underbrace{[a \quad b] \times \mathbf{z}_t}_{\tilde{\mathbf{z}}_t} = [a \quad b] \times \begin{bmatrix} 1 \\ \alpha \end{bmatrix} \times [a \quad b]\,\mathbf{z}_{t-1} + [a \quad b] \times \epsilon_t \tag{13}$$

$$= (a + b\alpha)\underbrace{[a \quad b] \times \mathbf{z}_{t-1}}_{\tilde{\mathbf{z}}_{t-1}} + \underbrace{[a \quad b] \times \epsilon_t}_{\tilde{\epsilon}_t}. \tag{14}$$

Hence, the two-dimensional VAR process reduces to a single linear AR process with $\tilde{\mathbf{z}}_t = [a \quad b] \times \mathbf{z}_t = az_{1t} + bz_{2t}$ and $\tilde{\epsilon}_t = [a \quad b] \times \epsilon_t = a\epsilon_{1t} + b\epsilon_{2t}$, which is a linear combination of the original two processes. In this case, we cannot recover $\mathbf{z}_t$ at all, but only the linear combination $\tilde{\mathbf{z}}_t$.

In summary, when the state transition matrices are not of full rank, there exist low-dimensional projections of the underlying latent processes that satisfy the observational equivalence everywhere. By assuming the nonsingular state transitions, one could prevent from recovering low-dimensional projections of latent causal factors and time-delayed relations.

### A.3 Proof of Identifiability Theory

#### A.3.1 Preliminaries

**Equivalent Relations on Latent Space**  Our proof of identifiability starts from deriving relations on estimated latent space from observational equivalence: the joint distribution $p_{\hat{g},\hat{f},\hat{p}_\epsilon}(\mathbf{x}_{\mathrm{Hx}}, \mathbf{x}_t)$ matches $p_{g,f,p_\epsilon}(\mathbf{x}_{\mathrm{Hx}}, \mathbf{x}_t)$ everywhere. Note that we consider only one future time step $\mathbf{x}_t$ for simplicity as the joint probability of the whole sequence can be decomposed into product of these terms. Since the learned mixing function $\mathbf{x}_t = \hat{g}(\mathbf{z}_t)$ can be written as $\mathbf{x}_t = (g \circ (g)^{-1} \circ \hat{g})(\mathbf{z}_t)$ because of injective properties of $(g, \hat{g})$, we can see that $\hat{g} = g \circ ((g)^{-1} \circ \hat{g}) = g \circ h$ for some function $h = (g)^{-1} \circ \hat{g}$ on the latent space. Our goal here is really to show that this function $h$, which represents the indeterminancy of the learned latent space, is a permutation with component-wise nonlinearities. It has been proved in (Klindt et al., 2020) that:

1. Indeterminancy $h$ on latent space can only be a bijection on the latent space if both $g$ and $\hat{g}$ are injective functions, and $h$ preserves the prior distribution in the latent space. The proofs are in Appendix A.1 of (Klindt et al., 2020) on Page 18;

2. Eq. 15 can be directly derived from observational equivalence using the injective properties of $(g, \hat{g})$. The proofs are in Appendix A.1 of (Klindt et al., 2020) on Page 19:

$$p(\mathbf{z}_t | \{\mathbf{z}_{t-\tau}\}) = p(h^{-1}(\mathbf{z}_t) | \{h^{-1}(\mathbf{z}_{t-\tau})\}) \frac{p(\mathbf{z}_t)}{p(h^{-1}(\mathbf{z}_t))} \quad \forall (\mathbf{z}_t, \{\mathbf{z}_{t-\tau}\}). \tag{15}$$

**Identifiability of Linear Non-Gaussian State-Space Model**  Linear State Space Model (SSM) defined below has been proved to be fully identifiable in (Zhang & Hyvärinen, 2011) when both the process noise $\epsilon_t$ and measurement error $\mathbf{e}_t$ are temporally white and independent of each other, and at most one component of the process noise $\epsilon_t$ is Gaussian. The observation error $\mathbf{e}_t$ can be either Gaussian or non-Gaussian:

$$\mathbf{x}_t = \mathbf{A}\mathbf{z}_t + \mathbf{e}_t, \tag{16}$$

$$\mathbf{z}_t = \sum_{\tau=1}^{L} \mathbf{B}_\tau \mathbf{z}_{t-\tau} + \epsilon_t. \tag{17}$$

We will make use of this property of linear non-Gaussian SSM for deriving **Theorem 2**. The main idea is if we can prove $h$ of parametric conditions (which also has vector autoregressive processes as in Eq. 17 with non-Gaussian noise) is within affine transformations, the componentwise identifiability of true latent variables can be directly derived because we can treat the affine indeterminacy as a "high-level" affine mixing of sources same as $\mathbf{A}$ in Eq. 16 without measurement error.

A.3.2 PROOF OF THEOREM 1

**Theorem A.1 (Nonparametric Processes)** *Assume nonparametric processes in Eq. 18, where the transition functions $f_i$ are third-order differentiable functions and mixing function $g$ is injective and differentiable almost everywhere; let $\mathbf{Pa}(z_{it})$ denote the set of (time-delayed) parent nodes of $z_{it}$:*

$$\underbrace{\mathbf{x}_t = g(\mathbf{z}_t)}_{\text{Nonlinear mixing}}, \quad \underbrace{z_{it} = f_i\left(\{z_{j,t-\tau}|z_{j,t-\tau} \in \mathbf{Pa}(z_{it})\}, \epsilon_{it}\right)}_{\text{Nonparametric transition}} \; with \; \underbrace{\epsilon_{it} \sim p_{\epsilon_i|\mathbf{u}}}_{\text{Nonstationary noise}} \; . \tag{18}$$

*Here we assume:*

1. *(Nonstationary Noise): Noise distribution $p_{\epsilon_i|\mathbf{u}}$ is modulated (in any way) by the observed categorical auxiliary variables $\mathbf{u}$, which denotes nonstationary regimes or domain index;*

2. *(Independent Noise): The noise terms $\epsilon_{it}$ are mutually independent (i.e., spatially and temporally independent) in each regime of $\mathbf{u}$ (note that this directly implies that $\epsilon_{it}$ are independent from $\mathbf{Pa}(z_{it})$ in each regime);*

3. *(Sufficient Variability): For any $\mathbf{z}_t \in \mathbb{R}^n$ there exist $2n + 1$ values for $\mathbf{u}$, i.e., $\mathbf{u}_j$ with $j = 0, 1, ..., 2n$, such that the $2n$ vectors $\mathbf{w}(\mathbf{z}_t, \mathbf{u}_{j+1}) - \mathbf{w}(\mathbf{z}_t, \mathbf{u}_j)$, with $j = 0, 1, ..., 2n$, are linearly independent with $\mathbf{w}(\mathbf{z}_t, \mathbf{u})$ defined below, where $q_i$ is the log density of the conditional distribution and $\mathbf{z}_{Hx} = \{\mathbf{z}_{t-\tau}\}$ denotes history information up to maximum time lag $L$:*

$$\mathbf{w}(\mathbf{z}_t, \mathbf{u}) \triangleq \left(\frac{\partial q_1(z_{1t}|\mathbf{z}_{Hx}, \mathbf{u})}{\partial z_{1t}}, \cdots, \frac{\partial q_n(z_{nt}|\mathbf{z}_{Hx}, \mathbf{u})}{\partial z_{nt}}, \frac{\partial^2 q_1(z_{1t}|\mathbf{z}_{Hx}, \mathbf{u})}{\partial z_{1t}^2}, \cdots, \frac{\partial^2 q_n(z_{nt}|\mathbf{z}_{Hx}, \mathbf{u})}{\partial z_{nt}^2}\right). \tag{19}$$

*Then the componentwise identifiability property of temporally causal latent processes is ensured.*

*Proof*: We first extend Eq. 15 to include conditioning on the nonstationary regime $\mathbf{u}$. We then show that if sufficient variability condition is satisfied, the indeterminancy function $h$ can only be permutation with component-wise nonlinearities.

**Step 1** We first derive equivalent relations on the latent space by conditioning on the nonstationary regime $\mathbf{u}$. This can be directly achieved by applying the change of variable formula on the $L + 1$ invertible maps: $\mathbf{z}_t \Rightarrow h^{-1}(\mathbf{z}_t), \mathbf{z}_{t-1} \Rightarrow h^{-1}(\mathbf{z}_{t-1}), ..., \mathbf{z}_{t-L} \Rightarrow h^{-1}(\mathbf{z}_{t-L})$. W.l.o.g, let's assume $L = 1$ for now. We then have the following three equalities:

$$p(\mathbf{z}_t, \mathbf{z}_{t-1}, \mathbf{u}) = p(h^{-1}(\mathbf{z}_t), h^{-1}(\mathbf{z}_{t-1}), \mathbf{u}) \left|\det \frac{\partial h^{-1}(\mathbf{z}_t)}{\partial \mathbf{z}_t}\right| \left|\det \frac{\partial h^{-1}(\mathbf{z}_{t-1})}{\partial \mathbf{z}_{t-1}}\right|, \tag{20}$$

$$p(\mathbf{z}_t) = p(h^{-1}(\mathbf{z}_t)) \left|\det \frac{\partial h^{-1}(\mathbf{z}_t)}{\partial \mathbf{z}_t}\right|, \tag{21}$$

$$p(\mathbf{z}_{t-1}, \mathbf{u}) = p(h^{-1}(\mathbf{z}_{t-1}), \mathbf{u}) \left|\det \frac{\partial h^{-1}(\mathbf{z}_{t-1})}{\partial \mathbf{z}_{t-1}}\right|. \tag{22}$$

Solving for the determinant terms in Eq. 21 and Eq. 22 and plugging them into Eq. 20, we have:

$$p(\mathbf{z}_t|\mathbf{z}_{t-1}, \mathbf{u}) = p(h^{-1}(\mathbf{z}_t)|h^{-1}(\mathbf{z}_{t-1}), \mathbf{u}) \frac{p(\mathbf{z}_t)}{p(h^{-1}(\mathbf{z}_t))}. \tag{23}$$

It is straightforward to see that this relation holds for multiple time lags where $L > 1$. We take logs on both sides, and we now define $\bar{q}(\mathbf{z}_t) \triangleq q(\mathbf{z}_t)$ as the marginal log-density of the components $\mathbf{z}_t$ when $\mathbf{u}$ is integrated out. We then have:

$$q(\mathbf{z}_t|\{\mathbf{z}_{t-\tau}\}, \mathbf{u}) - q(h^{-1}(\mathbf{z}_t)|h^{-1}(\{\mathbf{z}_{t-\tau}\}, \mathbf{u})) = \bar{q}(\mathbf{z}_t) - \bar{q}(h^{-1}(\mathbf{z}_t)), \tag{24}$$

and using the Independent Noise (IN) assumption, the conditional log-pdf $q(\mathbf{z}_t|\{\mathbf{z}_{t-\tau}\}, \mathbf{u})$ and its estimated version $q(h^{-1}(\mathbf{z}_t)|\{h^{-1}(\mathbf{z}_{t-\tau})\}, \mathbf{u})$ are conditional independent (note this has been enforced in causal process network as constraints) and LHS can be factorized as:

$$\sum_i q_i(z_{it}|\{\mathbf{z}_{t-\tau}\}, \mathbf{u}) - q_i\left(\left[h^{-1}(\mathbf{z}_t)\right]_i|\{h^{-1}(\mathbf{z}_{t-\tau})\}, \mathbf{u}\right) = \bar{q}(\mathbf{z}_t) - \bar{q}(h^{-1}(\mathbf{z}_t)). \quad (25)$$

where $\bar{q}$ is the marginal log-density of the components $\mathbf{z}_t$ when $\mathbf{u}$ is integrated out and it does not need to be factorial.

**Step 2** Now we do the following simplification of notations. Let $h_i^{-1}(\mathbf{z}_t) = \left[h^{-1}(\mathbf{z}_t)\right]_i$. Denote the first-order and second-order derivatives by a superscript as:

$$q_i^1(z_{it}|\{\mathbf{z}_{t-\tau}\}, \mathbf{u}) = \frac{\partial q_i(z_{it}|\{\mathbf{z}_{t-\tau}\}, \mathbf{u})}{\partial z_{it}}, \quad (26)$$

$$q_i^2(z_{it}|\{\mathbf{z}_{t-\tau}\}, \mathbf{u}) = \frac{\partial^2 q_i(z_{it}|\{\mathbf{z}_{t-\tau}\}, \mathbf{u})}{\partial z_{it}^2}, \quad (27)$$

and take derivatives of both sides of Eq. 25 with respect to $z_{jt}$, we have:

$$q_j^1(z_{jt}|\{\mathbf{z}_{t-\tau}\}, \mathbf{u}) - \sum_{i=1}^n q_i^1(h_i^{-1}(\mathbf{z}_t)|\{h^{-1}(\mathbf{z}_{t-\tau})\}, \mathbf{u})\frac{\partial h_i^{-1}(\mathbf{z}_t)}{\partial z_{jt}} \quad (28)$$

$$= \bar{q}^j(\mathbf{z}_t) - \sum_i \bar{q}^j(h_i^{-1}(\mathbf{z}_t))\frac{\partial h_i^{-1}(\mathbf{z}_t)}{\partial z_{jt}}. \quad (29)$$

Denote the first order derivative of $h^{-1}$ as $v_i^j(\mathbf{z}_t) = \frac{\partial h_i^{-1}(\mathbf{z}_t)}{\partial z_{jt}}$ and $v_i^{jj'}(\mathbf{z}_t)$ is the second-order derivative with respect to a different component $z_{j't}$ for any $j \neq j'$. Taking another derivative with respect to $z_{j't}$ on both sides of Eq. 29, the first term on LHS vanishes and we have:

$$\sum_i q_i^{11}(h_i^{-1}(\mathbf{z}_t)|\{h^{-1}(\mathbf{z}_{t-\tau})\}, \mathbf{u})v_i^j(\mathbf{z}_t)v_i^{j'}(\mathbf{z}_t) + q_i^1(h_i^{-1}(\mathbf{z}_t)|\{h^{-1}(\mathbf{z}_{t-\tau})\}, \mathbf{u})v_i^{jj'}(\mathbf{z}_t) = c^{jj'}. \quad (30)$$

where $c^{jj'}$ denotes the derivatives of RHS of Eq. 29 which **does not depend on** $\mathbf{u}$. Same as (Hyvarinen et al., 2019), we collect all these equations in vector form by defining $\mathbf{a}_i(y)$ as a vector collecting all entries $v_i^j(\mathbf{z}_t)v_i^{j'}(\mathbf{z}_t)$ for $j \in [1, n]$ and $j' \in [1, j-1]$. We omit diagonal terms, and by symmetry, take only one half of the indices. Likewise, collect all the entries $v_i^{jj'}(\mathbf{z}_t)$ for $j \in [1, n]$ and $j' \in [1, j-1]$ in the vector $\mathbf{b}(\mathbf{z}_t)$. All the entries of $c^{jj'}$ are in $\mathbf{c}(\mathbf{z}_t)$. These $n(n-1)/2$ equations can be written a single system of equations:

$$\sum_i \mathbf{a}_i(y)q_i^{11}(h_i^{-1}(\mathbf{z}_t)|\{h^{-1}(\mathbf{z}_{t-\tau})\}, \mathbf{u}) + \mathbf{b}_i(\mathbf{z}_t)q_i^1(h_i^{-1}(\mathbf{z}_t)|\{h^{-1}(\mathbf{z}_{t-\tau})\}, \mathbf{u}) = \mathbf{c}(\mathbf{z}_t). \quad (31)$$

Now, collect the $\mathbf{a}$ and $\mathbf{b}$ into a matrix $\mathbf{M}$:

$$\mathbf{M}(\mathbf{z}_t) = (\mathbf{a}_1(\mathbf{z}_t), \ldots, \mathbf{a}_n(\mathbf{z}_t), \mathbf{b}_i(\mathbf{z}_t), \ldots, \mathbf{b}_n(\mathbf{z}_t)). \quad (32)$$

Eq. 31 takes the form of the following linear system:

$$\mathbf{M}(\mathbf{z}_t)\mathbf{w}(\mathbf{z}_t, \mathbf{u}) = \mathbf{c}(\mathbf{z}_t), \quad (33)$$

where $\mathbf{w}$ are the vectors defined in the sufficient variability assumption, and $\mathbf{w}$ is defined for any input $\mathbf{z}_t$. Notice that the RHS of the linear system does not depend on $\mathbf{u}$, so we fix $\mathbf{z}_t$ and consider the $2n+1$ points $\mathbf{u}$ given for that $\mathbf{z}_t$ by the sufficient variability assumption.

Collect Eq. 33 above for $2n$ points starting from index 1:

$$\mathbf{M}(\mathbf{z}_t)(\mathbf{w}(\mathbf{z}_t, \mathbf{u}_1), \ldots, \mathbf{w}(\mathbf{z}_t, \mathbf{u}_{2n})) = (\mathbf{c}(\mathbf{z}_t), \ldots, \mathbf{w}(\mathbf{z}_t, \mathbf{u}_1)), \quad (34)$$

and collect the equation starting from index 0 for $2n$ points:

$$\mathbf{M}(\mathbf{z}_t)\left(\mathbf{w}(\mathbf{z}_t, \mathbf{u}_0), \dots, \mathbf{w}(\mathbf{z}_t, \mathbf{u}_{2n-1})\right) = \left(\mathbf{c}(\mathbf{z}_t), \dots, \mathbf{w}(\mathbf{z}_t, \mathbf{u}_1)\right). \tag{35}$$

Substract Eq. 35 from Eq. 34, we then have:

$$\mathbf{M}(\mathbf{z}_t)\underbrace{\left[\mathbf{w}(\mathbf{z}_t, \mathbf{u}_1) - \mathbf{w}(\mathbf{z}_t, \mathbf{u}_0), \dots, \mathbf{w}(\mathbf{z}_t, \mathbf{u}_{2n}) - \mathbf{w}(\mathbf{z}_t, \mathbf{u}_0)\right]}_{\mathbf{W}} = \mathbf{0}. \tag{36}$$

By the suffienct variability assumption, the matrix $\mathbf{W}$ that has linearly independent columns and is a square matrix so is nonsingular. The only solution to the linear system above is thus:

$$\mathbf{M}(\mathbf{z}_t) = \left(\mathbf{a}_1(\mathbf{z}_t), \dots, \mathbf{a}_n(\mathbf{z}_t), \mathbf{b}_i(\mathbf{z}_t), \dots, \mathbf{b}_n(\mathbf{z}_t)\right) = \mathbf{0}. \tag{37}$$

Following (Hyvarinen et al., 2019), $\mathbf{a}(\mathbf{z}_t)$ being zero implies no row of the Jacobian of $h^{-1}(\mathbf{z}_t)$ can have more than one non-zero entry. This holds for any $\mathbf{z}_t$. By continuity of the Jacobian and its invertibility, the non-zero entries in the Jacobian must be in the same places for all $\mathbf{z}_t$: If they switched places, there would have to be a point where the Jacobian is singular, which would contradict the bijection properties of $h^{-1}$ derived in Section A.3.1. This means that each $h_i^{-1}(\mathbf{z}_t)$ is a function of only one $z_{kt}$ for $k \in [1, n]$. The bijection $h^{-1}$ also implies that each of the componentwise functions is invertible. Thus, we have proven that latent variables are identifiable up to permutation and componentwise invertible transformations and temporally causal latent processes with conditions required by **Theorem 1** are proved to be identifiable from observed variables. ∎

### A.3.3 PROOF OF THEOREM 2

**Theorem A.2 (Parametric Processes)** *Assume the vector autoregressive process in Eq. 38, where the state transition functions are* linear and additive *and mixing function $g$ is injective and differentiable almost everywhere. Let $\mathbf{B}_\tau \in \mathbb{R}^{n \times n}$ be the state transition matrix at lag $\tau$. The process noises $\epsilon_{it}$ are assumed to be stationary and both spatially and temporally independent:*

$$\underbrace{\mathbf{x}_t = g(\mathbf{z}_t)}_{\text{Nonlinear mixing}}, \quad \underbrace{\mathbf{z}_t = \sum_{\tau=1}^{L} \mathbf{B}_\tau \mathbf{z}_{t-\tau} + \epsilon_t}_{\text{Linear additive transition}} \text{ with } \underbrace{\epsilon_{it} \sim p_{\epsilon_i}}_{\text{Independent noise}}. \tag{38}$$

*Here we assume:*

1. *(Generalized Laplacian Noise): Process noises $\epsilon_{it} \sim p_{\epsilon_i}$ are mutually independent and follow the generalized Laplacian distribution $p_{\epsilon_i} = \frac{\alpha_i \lambda_i}{2\Gamma(1/\alpha_i)} \exp\left(-\lambda_i |\epsilon_i|^{\alpha_i}\right)$ with $\alpha_i < 2$;*

2. *(Nonsingular State Transitions): For at least one $\tau$, the state transition matrix $\mathbf{B}_\tau$ is of full rank.*

*Then the componentwise identifiability property of temporally causal latent processes is ensured.*

*Proof*: The following proof is inspired by Theorem 1 in (Klindt et al., 2020). The key differences are (i) allowing temporal causal relations $\mathbf{B}_\tau$ among the sources instead of independent sources assumption, (ii) extending the single time lag restriction to multiple time lags case.

**Identifiability on Causally-Related Sources** Let us start from the simple case where the time lag $\tau = 1$. In this case, the transition dynamic in Eq. 38 can be simplified as

$$\mathbf{x}_t = g(\mathbf{z}_t), \quad \mathbf{z}_t = \mathbf{B}\mathbf{z}_{t-1} + \epsilon_t. \tag{39}$$

Using Eq. 15 and by applying the distributional forms of generalized Laplacian noise, we have:

$$p(\mathbf{z}_t | \mathbf{z}_{t-1}) = p(h^{-1}(\mathbf{z}_t) | h^{-1}(\mathbf{z}_{t-1})) \frac{p(\mathbf{z}_t)}{p(h^{-1}(\mathbf{z}_t))}$$

$$\implies M||\mathbf{z}_t - \mathbf{B}\mathbf{z}_{t-1}||_\alpha^\alpha - N||h^{-1}(\mathbf{z}_t) - \mathbf{B}h^{-1}(\mathbf{z}_{t-1})||_\alpha^\alpha = \log \frac{p(\mathbf{z}_t)}{p(h^{-1}(\mathbf{z}_t))}, \tag{40}$$

where $M$ and $N$ are the constants appearing in the exponentials in $p(\mathbf{z}_t|\mathbf{z}_{t-1})$ and $p(h^{-1}(\mathbf{z}_t)|h^{-1}(\mathbf{z}_{t-1}))$.

Taking the derivative w.r.t $\mathbf{z}_{t-1}$ on both sides, we obtain

$$\frac{\partial||\mathbf{z}_t - \mathbf{B}\mathbf{z}_{t-1}||_\alpha^\alpha}{\partial\mathbf{z}_{t-1}} = \frac{\partial||h^{-1}(\mathbf{z}_t) - \mathbf{B}h^{-1}(\mathbf{z}_{t-1})||_\alpha^\alpha}{\partial\mathbf{z}_{t-1}}. \tag{41}$$

For the left hand of the Eq. 41, we can derive

$$\frac{\partial||\mathbf{z}_t - \mathbf{B}\mathbf{z}_{t-1}||_\alpha^\alpha}{\partial||\mathbf{z}_t - \mathbf{B}\mathbf{z}_{t-1}||_\alpha}\frac{\partial||\mathbf{z}_t - \mathbf{B}\mathbf{z}_{t-1}||_\alpha}{\partial\mathbf{z}_{t-1}} = \frac{\partial||h^{-1}(\mathbf{z}_t) - \mathbf{B}h^{-1}(\mathbf{z}_{t-1})||_\alpha^\alpha}{\partial\mathbf{z}_{t-1}}$$

$$\implies \quad \alpha||\mathbf{z}_t - \mathbf{B}\mathbf{z}_{t-1}||_\alpha^{\alpha-1}\frac{\partial||\mathbf{z}_t - \mathbf{B}\mathbf{z}_{t-1}||_\alpha}{\partial\mathbf{z}_{t-1}} = \frac{\partial||h^{-1}(\mathbf{z}_t) - \mathbf{B}h^{-1}(\mathbf{z}_{t-1})||_\alpha^\alpha}{\partial\mathbf{z}_{t-1}}$$

$$\implies \quad \alpha||\mathbf{z}_t - \mathbf{B}\mathbf{z}_{t-1}||_\alpha^{\alpha-1}\frac{\partial||\epsilon_t||_\alpha}{\partial\epsilon_t}\frac{\partial\mathbf{z}_t - \mathbf{B}\mathbf{z}_{t-1}}{\partial\mathbf{z}_{t-1}} = \frac{\partial||h^{-1}(\mathbf{z}_t) - \mathbf{B}h^{-1}(\mathbf{z}_{t-1})||_\alpha^\alpha}{\partial\mathbf{z}_{t-1}} \tag{42}$$

$$\implies \quad -\alpha||\mathbf{z}_t - \mathbf{B}\mathbf{z}_{t-1}||_\alpha^{\alpha-1}\frac{\partial||\epsilon_t||_\alpha}{\partial\epsilon_t}\mathbf{B} = \frac{\partial||h^{-1}(\mathbf{z}_t) - \mathbf{B}h^{-1}(\mathbf{z}_{t-1})||_\alpha^\alpha}{\partial\mathbf{z}_{t-1}}$$

$$\implies \quad -\alpha(\mathbf{z}_t - \mathbf{B}\mathbf{z}_{t-1}) \odot |\mathbf{z}_t - \mathbf{B}\mathbf{z}_{t-1}|^{\alpha-2}\mathbf{B} = \frac{\partial||h^{-1}(\mathbf{z}_t) - \mathbf{B}h^{-1}(\mathbf{z}_{t-1})||_\alpha^\alpha}{\partial\mathbf{z}_{t-1}}.$$

Making the same derivation process on the right hand, we obtain

$$- \alpha(\mathbf{z}_t - \mathbf{B}\mathbf{z}_{t-1}) \odot |\mathbf{z}_t - \mathbf{B}\mathbf{z}_{t-1}|^{\alpha-2}\mathbf{B}$$
$$= - \alpha(h^{-1}(\mathbf{z}_t) - \mathbf{B}h^{-1}(\mathbf{z}_{t-1})) \odot |h^{-1}(\mathbf{z}_t) - \mathbf{B}h^{-1}(\mathbf{z}_{t-1})|^{\alpha-2}\mathbf{B}\frac{\partial h^{-1}(\mathbf{z}_{t-1})}{\mathbf{z}_{t-1}}. \tag{43}$$

For any $\mathbf{z}_t$ we can choose $\mathbf{z}_t = \mathbf{B}\mathbf{z}_{t-1}$ and thus the Eq. 43 can be written as:

$$(h^{-1}(\mathbf{z}_t) - \mathbf{B}h^{-1}(\mathbf{z}_{t-1})) \odot |h^{-1}(\mathbf{z}_t) - \mathbf{B}h^{-1}(\mathbf{z}_{t-1})|^{\alpha-2}\mathbf{B}\frac{\partial h^{-1}(\mathbf{z}_{t-1})}{\mathbf{z}_{t-1}} = \mathbf{0}. \tag{44}$$

Considering the nonsingularity of matrix $\mathbf{B}$ (by assumption) and bijection of $h$, we can derive

$$(h^{-1}(\mathbf{z}_t) - \mathbf{B}h^{-1}(\mathbf{z}_{t-1})) \odot |h^{-1}(\mathbf{z}_t) - \mathbf{B}h^{-1}(\mathbf{z}_{t-1})|^{\alpha-2} = \mathbf{0}$$
$$\implies \quad (h_i^{-1}(\mathbf{z}_t) - \mathbf{B}h_i^{-1}(\mathbf{z}_{t-1}))|h_i^{-1}(\mathbf{z}_t) - \mathbf{B}h_i^{-1}(\mathbf{z}_{t-1})|^{\alpha-2} = 0. \tag{45}$$

for all $i = 1, \ldots, d$. Apparently $h^{-1}(\mathbf{z}_t) = \mathbf{B}h^{-1}(\mathbf{z}_{t-1})$ is the only solution, thus

$$h^{-1}(\mathbf{B}\mathbf{z}_{t-1}) = \mathbf{B}h^{-1}(\mathbf{z}_{t-1}). \tag{46}$$

Substitute Eq. 46 to the right hand of Eq. 40, we have

$$||z_t - \mathbf{B}z_{t-1}||_\alpha^\alpha = ||h^{-1}(z_t) - \mathbf{B}h^{-1}(z_{t-1})||_\alpha^\alpha$$
$$\implies \quad ||z_t - \mathbf{B}z_{t-1}||_\alpha^\alpha = ||h^{-1}(z_t) - h^{-1}(\mathbf{B}z_{t-1})||_\alpha^\alpha. \tag{47}$$

This indicates that $h^{-1}$ preserves the $\alpha$-distances between points. Since $h$ is bijective, then by Mazur-Ulam theorem Mazur & Ulam (1932), $h$ must be an affine transform. According to Theorem 2 in (Zhang & Hyvärinen, 2011), the model is identifiable, which proves the theorem.

**Extension to Multiple Time Lags** We can extend the result in Eq. 41 and have

$$\frac{\partial||\mathbf{z}_t - \sum_{\tau=1}^L \mathbf{B}_\tau\mathbf{z}_{t-\tau}||_\alpha^\alpha}{\partial\mathbf{z}_{t-i}} = \frac{\partial||h^{-1}(\mathbf{z}_t) - \sum_{\tau=1}^L \mathbf{B}_\tau h^{-1}(\mathbf{z}_{t-\tau})||_\alpha^\alpha}{\partial\mathbf{z}_{t-i}}, \tag{48}$$

where $\mathbf{z}_{t-i}$ is any lag latents with the limitation that the $\mathbf{B}_i$ corresponding to $\mathbf{z}_{t-i}$ is of full rank. Following the same above-mentioned derivation process, we can obtain

$$(\mathbf{z}_t - \sum_{\tau=1}^L \mathbf{B}_\tau\mathbf{z}_{t-\tau}) \odot |\mathbf{z}_t - \sum_{\tau=1}^L \mathbf{B}_\tau\mathbf{z}_{t-\tau}|^{\alpha-2}\mathbf{B}_i$$
$$= (h^{-1}(\mathbf{z}_t) - \sum_{\tau=1}^L \mathbf{B}_\tau h^{-1}(\mathbf{z}_{t-\tau})) \odot |h^{-1}(\mathbf{z}_t) - \sum_{\tau=1}^L \mathbf{B}_\tau h^{-1}(\mathbf{z}_{t-\tau})|^{\alpha-2}\mathbf{B}_i\frac{\partial h^{-1}(\mathbf{z}_{t-i})}{\mathbf{z}_{t-i}}. \tag{49}$$

For any $\mathbf{z}_t$ we can choose $\mathbf{z}_t = \sum_{\tau=1}^{L} \mathbf{B}_\tau \mathbf{z}_{t-\tau}$, thus the Eq. 49 can be written as:

$$(h^{-1}(\mathbf{z}_t) - \sum_{\tau=1}^{L} \mathbf{B}_\tau h^{-1}(\mathbf{z}_{t-\tau})) \odot |h^{-1}(\mathbf{z}_t) - \sum_{\tau=1}^{L} \mathbf{B}_\tau h^{-1}(\mathbf{z}_{t-\tau})|^{\alpha-2} \mathbf{B}_i \frac{\partial h^{-1}(\mathbf{z}_{-i})}{\mathbf{z}_{t-i}} = \mathbf{0}. \quad (50)$$

As mentioned above, $h^{-1}(\mathbf{z}_t) = \sum_{\tau=1}^{L} \mathbf{B}_\tau h^{-1}(\mathbf{z}_{t-\tau})$ is the only solution, thus

$$h^{-1}(\sum_{\tau=1}^{L} \mathbf{B}_\tau \mathbf{z}_{t-\tau}) = \sum_{\tau=1}^{L} \mathbf{B}_\tau h^{-1}(\mathbf{z}_{t-\tau}). \quad (51)$$

Following the same procedure in the simple case, the theorem is proven. ∎

### A.4 COMPARISONS WITH EXISTING THEORIES

The closest work to ours includes (1) PCL (Hyvarinen & Morioka, 2017), which exploited temporal constraints to separate independent sources, (2) SlowVAE (Klindt et al., 2020), which leveraged sparse transition of adjacent video frames to separate independent sources, and (3) iVAE (Khemakhem et al., 2020), which leveraged the nonstationarity by the modulation of side information $\mathbf{u}$ on the prior distribution $p(\mathbf{z}|\mathbf{u})$ of conditional factorial latent variables. Our work extends the theories to the discovery of the conditional independent sources with time-delayed causal relations in between by leveraging nonstationarity, or functional and distribution forms of temporal statistics. To the best of our knowledge, this is one of the first works that successfully recover time-delayed latent processes from their nonlinear mixtures without using sparsity or minimality assumptions.

**PCL** The sources $z_{it}$ in PCL were assumed to be mutually independent (see Assumption 1 of Theorem 1 in PCL). In contrast, we allow the sources to have time-delayed causal relations in between, which is much more realistic in real-world applications. They further assumed the sources are stationary, while we allow nonstationarity in the nonparametric setting (the nonstationary noise assumption). The underlying processes of PCL are described by Eq. 52:

$$\log p(z_{i,t}|z_{i,t-1}) = G(z_{i,t} - \rho z_{i,t-1}) \quad \text{or} \quad \log p(z_{i,t}|z_{i,t-1}) = -\lambda (z_{i,t} - r(z_{i,t-1}))^2 + \text{const}. \quad (52)$$

where $G$ is some non-quadratic function corresponding to the log-pdf of innovations, $\rho < 1$ is regression coefficient, $r$ is some nonlinear, strictly monotonic regression, and $\lambda$ is a positive precision parameter. Both theorems of our work extend the theory to the discovery of the conditional independent sources with time-delayed causal relations in between. Furthermore, in the nonparametric process condition, we do not restrict the functional and distributional forms of underlying transitions. Our proposed nonparametric condition naturally includes PCL as a special case.

**SlowVAE** Inspired by slow feature analysis, SlowVAE assumes the underlying sources to have identity transitions with generalized Laplacian innovations described in Eq. 53:

$$p(\mathbf{z}_t|\mathbf{z}_{t-1}) = \prod_{i=1}^{d} \frac{\alpha\lambda}{2\Gamma(1/\alpha)} \exp -(\lambda|z_{i,t} - z_{i,t-1}|^\alpha) \quad with \quad \alpha < 2. \quad (53)$$

Our proposed parametric condition (Theorem 2) in Eq. 4 is a natural extension to the Laplacian innovation model above by allowing time-delayed vector autoregressive transitions in the latent process with multiple time lags. Consequently, temporally causally-related latent processes with linear transition dynamics can thus be modeled and recovered from their nonlinear mixtures with our parametric condition.

**iVAE** Similar to TCL (Hyvarinen & Morioka, 2016) and GIN (Sorrenson et al., 2020), iVAE exploits the nonstationarity brought by the side information (i.e., class label) on the prior distribution of latent variables $\mathbf{z}_t$. As one can see from Eq. 54, the latent variables are conditionally independent, without causal relations in between while both of our theorems consider (time-delayed) causal relations between latent variables. In addition, iVAE exploits the nonstationarity brought by side information (i.e., class label) on the prior distribution of latent variables $\mathbf{z}$. On the contrary, our nonparametric condition, instead of relying on the change in the prior distribution of latent variables,

exploits the nonstationarity in the noise distribution, which is more natural in real-world datasets. Finally, iVAE assumes modulated exponential families in Eq. 54 while our nonparametric condition (Theorem 1) allows any kinds of modulation by side information $\mathbf{u}$ without those strong assumptions on the transition functions or distributions.

$$p_{T,\lambda}(\mathbf{z}|\mathbf{u}) = \prod_i \frac{Q_i(z_i)}{Z_i(\mathbf{u})} \exp\left[\sum_{j=1}^k T_{i,j}(z_i)\lambda_{i,j}(\mathbf{u})\right] \tag{54}$$

In terms of architecture innovations, to remove distributional and functional form constraints of iVAE, we design a novel causal transition prior network for nonparametric transitions by injecting the IN condition inside reparameterization trick, resulting in an efficient scoring mechanism of transition prior which only needs to compute the determinant of a low-triangular Jacobian matrix. This module was never seen in previous work.

### A.5 CONNECTING THEORIES TO MODEL

As one can see from the proofs in Appendix A.3, what have been assumed for the estimation framework are the conditional factorial properties of $q(\hat{\mathbf{z}}_t|\{\hat{\mathbf{z}}_{t-\tau}\}, \mathbf{u})$ where $\hat{\mathbf{z}}_t = h^{-1}(\mathbf{z}_t)$ and the model of temporal nonstationarities through nonstationary noises. The conditional factorial properties have been injected using the reparameterization trick (Eq. 7) with the IN condition in causal transition prior and the enforcing of spatiotemporal independence of estimated residuals through contrastive learning. The nonstationary noises are modeled with flow-based density estimators. We share the weights of the other modules (e.g., encoder, transition function, decoder, inference network, etc.) across nonstationary regimes while using separate flow models to estimate the density of residuals and evaluate the prior scores in each regime. We also use componentwise flow models so the learned residuals will not interact with each other in the estimation framework. Finally, in nonparametric processes, we warm-start the flow models to generate standard Gaussian noise. In parametric processes, the flow models are initialized to generate standard Laplacian noise. Note that the other assumed conditions in the two theorems, such as sufficient variability and nonsingular state transitions, are data properties and do not need to be encoded as constraints in the estimation framework.

## B EXPERIMENT SETTINGS

### B.1 SYNTHETIC DATASET

Seven synthetic datasets, including two datasets (NP and VAR) which satisfy our assumptions, and five datasets, which violate each of the assumption in the proposed theorems, are used in this paper. We set the latent size $n = 8$ and the lag number of the process $L = 2$. The mixing function $g$ is a random three-layer MLP with LeakyReLU units.

**Nonparametric (NP) Dataset** For nonparametric processes, we generate 150,000 data points according to Eq. 2. In particular, we use a Gaussian additive noise model as the latent processes. The noises $\epsilon_{it}$ are sampled from i.i.d. Gaussian distribution with variance modulated by 20 different nonstationary regimes. In each regime, the variance entries are uniformly sampled between 0 and 1. A 2-layer MLP with LeakyReLU units is used as the state transition function $f_i$. When we need sparse causal structure for visualization, a random binary mask is added to the input nodes.

**(Violation) Insufficient Variability** For this dataset, we create datasets that violate the nonstationary noise condition and sufficient variability by restricting the number of nonstationary regimes observed in the NP dataset. When only one regime is observed, we violate the nonstationary noise condition by using stationary noise. Furthermore, we vary the number of the observed regimes $|\mathbf{u}| \in \{1, 5, 10, 15, 20\}$ to assess the impacts of variability on the recovery of nonparametric processes.

**Parametric (VAR) Dataset** For parametric processes, we generate 50,000 data points according to Eq. 4. The noises $\epsilon_{it}$ are sampled from i.i.d. Laplacian distribution ($\sigma = 0.1$). The entries of state transition matrices $\mathbf{B}_\tau$ are uniformly distributed between $[-0.5, 0.5]$.

**(Violation) Low-rank State Transition**    For this dataset, the transition matrix $\mathbf{B}_\tau$ in Eq. 4 is low-rank instead of full-rank. The datasets are created following the steps in the VAR dataset, but we restrict the rank of state transition matrix $\mathbf{B}_\tau$ to 4 and time lag $L = 1$. The full matrix rank is 8.

**(Violation) Gaussian Noise Distribution**    For this dataset, the noise terms $\epsilon_{it}$ in Eq. 4 follow the Gaussian distribution ($\alpha_i = 2$) instead of Generalized Laplacian distribution ($\alpha_i < 2$). In particular, the noise terms $\epsilon_{it}$ are sampled from i.i.d. Gaussian distribution ($\sigma = 0.1$).

**(Violation) Regime-Variant Causal Relations**    For regime-variant causal relations, we generate 240,000 data points according to Eq. 55:

$$\mathbf{x}_t = g(\mathbf{z}_t), \quad \mathbf{z}_t = \sum_{\tau=1}^{L} \mathbf{B}_\tau^{\mathbf{u}} \mathbf{z}_{t-\tau} + \epsilon_t \quad with \quad \epsilon_{it} \sim p_{\epsilon_i}. \tag{55}$$

The noises $\epsilon_{it}$ are sampled from i.i.d. Laplace distribution ($\sigma = 0.1$). In each regime $\mathbf{u}$, the entries of state transition matrices $\mathbf{B}_\tau^{\mathbf{u}}$ are uniformly distributed between $[-0.5, 0.5]$.

**(Violation) Instantaneous Causal Relations**    For instantaneous causal relations, we generate 45,000 data points according to Eq. 56:

$$\mathbf{x}_t = g(\mathbf{z}_t), \quad \mathbf{z}_t = \mathbf{A}\mathbf{z}_t + \sum_{\tau=1}^{L} \mathbf{B}_\tau \mathbf{z}_{t-\tau} + \epsilon_t \quad with \quad \epsilon_{it} \sim p_{\epsilon_i}, \tag{56}$$

where matrix $\mathbf{A}$ is a random Directed Acyclic Graph (DAG) which contains the coefficients of the linear instantaneous relations. The noises $\epsilon_{it}$ are sampled from i.i.d. Laplacian distribution with $\sigma = 0.1$. The entries of state transition matrices $\mathbf{B}_\tau$ are uniformly distributed between $[-0.5, 0.5]$.

## B.2    REAL-WORLD DATASET

Three public datasets, including KiTTiMask, Mass-Spring System, and CMU MoCap database, are used. The observations together with the true temporally causal latent processes are showcased in Fig. B.1. For CMU MoCap, the true latent causal variables and time-delayed relations are unknown.

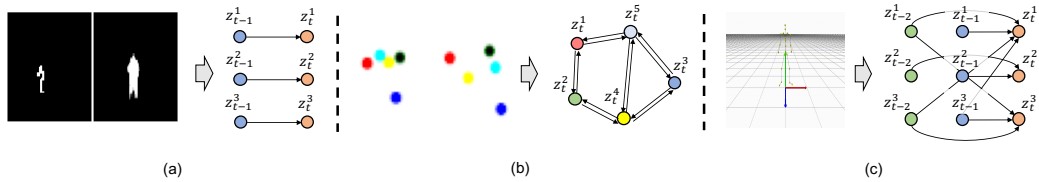

(a) (b) (c)

Figure B.1: Real-world datasets: (a) KiTTiMask is a video dataset of binary pedestrian masks, (b) Mass-Spring system is a video dataset with ball movement rendered in color and invisible springs, and (c) CMU MoCap is a 3D point cloud dataset of skeleton-based signals.

**KiTTiMask**    The KiTTiMask dataset consists of pedestrian segmentation masks sampled from the autonomous driving vision benchmark KiTTi-MOTS. For each given frame, the position (vertical and horizontal) and the scale of the pedestrian masks are set using measured values. The difference in the sample time (e.g., $\Delta t = 0.15s$) generates the sparse Laplacian innovations between frames.

**Mass-Spring System**    The Mass-Spring system is a classical physical system that several objects are connected by some visible/invisible spring, which follows Hooke's law. In this work, we considered the system with five degrees of freedom and made linearization on the state without calculating the Euclidian distance between objects. Thus, there are ten causal relations, six of which were set connected, and the other four were disconnected. The rest length of the spring was uniformly distributed between $[1, 10]$, and the stiffness of the spring relation was set as 20. The action was $a_t = 300e_t$, where $e_t$ followed the Laplacian distribution with mean $\mu = 0$ and variance $\sigma = 1$. We assumed there was no damping in the system and randomly assigned the objects in different positions at the beginning of each episode.

**CMU MoCap** CMU MoCap (http://mocap.cs.cmu.edu/) is an open-source human motion capture dataset with various motion capture recordings (e.g., walk, jump, basketball, etc.) performed by over 140 subjects. In this work, we fit our model on 12 trials of "walk" recordings (Subject 7). Skeleton-based measurements have 62 observed variables corresponding to the locations of joints (e.g., head, foot, shoulder, wrist, throat, etc.) of the human body at each time step.

### B.3 EVALUATION METRICS

**SHD: Structural Hamming Distance** We use SHD (de Jongh & Druzdzel, 2009) to measure the distance between two causal graphs. It computes the number of edge insertions, deletions, or flips in order to transform one graph to another graph. SHD is one variant of Minimum Edit Distance (MED) in causal discovery area by allowing only insertions, deletions, and flips of edges.

**MCC: Mean Correlation Coefficient** MCC is a standard metric for evaluating the recovery of latent factors in ICA literature. MCC first calculates the absolute values of correlation coefficient between every ground-truth factor against every estimated latent variable. Depending on whether componentwise invertible nonlinearities exist in the recovered factors, Pearson correlation coefficients or Spearman's rank correlation coefficients can be used. The possible permutation is adjusted by solving a linear sum assignment problem in polynomial time on the computed correlation matrix.

In this work, we use the Pearson correlation coefficient for the VAR processes and Spearman's correlation coefficient for the NP processes.

## C IMPLEMENTATION DETAILS

In this section, we first provide the network architecture details of LEAP. The hyperparameter selection criteria and sensitivity analysis results are presented. The training settings are summarized.

### C.1 NETWORK ARCHITECTURE

We summarize our network architecture below and describe it in detail in Table C.1 and Table C.2.

- **(1,2) MLP-Encoder and MLP-Decoder**: These modules are used for the synthetic and motion capture datasets. They are composed of a series of fully-connected neural networks with LeakyReLU as the activation function. The universal approximation theorem guarantees that our model can approximate the mixing function. The encoder maps the raw observations into features, while the decoder maps the latent variables back to the inputs.

- **(3,4) CNN-Encoder and CNN-Decoder**: For the KiTTiMask dataset, vanilla CNNs are used for both the encoder and decoder. For the Mass-Spring system dataset, the time-delayed causal variables use objects as the building blocks to factorize the scene. The object-centric representations contain object locations and some other attributes (e.g., color, size, etc.). We thus use two separate CNNs, one for extracting visual features (see Feature Extractor in Table C.2) and the other for locating object locations with a spatial softmax unit (see Keypoint Predictor in Table C.2). The decoder retrieves object features from feature maps using object locations and reconstructs the scene (see Refiner in Table C.2).

- **(5) Inference Network**: We apply the bidirectional Gated Recurrent Unit (GRU) (Cho et al., 2014) to preserve both the past and the future information. It processes the input sequence $\mathbf{x}_t$ in both directions: one for the forward pass and one for the backward pass. We denote the forward and backward embeddings as $\overrightarrow{H}$ and $\overleftarrow{H}$. The other inference module (see `TemporalDynamics` in Table C.1) uses the sampled/inferred past latent variables to compute the posterior of $\mathbf{z}_t$. We insert skip-connections (He et al., 2016) between the two inference modules to avoid the vanishing gradient problem and obtain better model convergence performances. Note that for the first $L$ temporally-earliest latent variables in the sequence, there is no time-delayed information, and we use isotropic Gaussian $\mathcal{N}(0, 1)$ as their prior distributions. The prior of the remaining sequence is evaluated with the learned transition prior network described below.

- **(6,7) Causal Process Prior Network** This module contains three components. (i) Inverse transition functions. For the NP transitions, we use MLPs to compute the estimated noises. For

the VAR transitions, a group of state transition matrices, one for each time lag, is used. (ii) $\log\left(|\det\left(\mathbf{J}\right)|\right)$ computation. For the NP transitions, the Jacobian matrix entries $\frac{\partial r_i}{\partial \hat{z}_{it}}$ are computed using `torch.autograd.functional.jacobian` method. The log determinant is then evaluated by summing over log transformations of absolute values of Jacobian terms for $\hat{z}_{it}$. For the VAR transitions, because of the additive noise assumption, the log determinant is directly 0. (iii) Spline flow model. Componentwise spline flow models use monotonic linear rational splines to transform standard Gaussian distribution to the estimated noise distribution. We use eight bins for the linear splines and set the bound as five so data points lying outside $[-5, 5]$ are evaluated using $\mathcal{N}(0, 1)$ directly while the data points within the region are evaluated by spline flow models. For the nonparametric processes, we always warm-start the spline flows by training it on a dataset of standard Gaussian noises with steps=5000 and learning rate=0.001. For the parametric processes, we warm-start it instead on a dataset of standard Laplacian noises. All the three components of the transition prior network are set to be learnable during the VAE updates.

Table C.1: Architecture details. BS: batch size, T: length of time series, i_dim: input dimension, z_dim: latent dimension, LeakyReLU: Leaky Rectified Linear Unit.

| Configuration | Description | Output |
|---|---|---|
| **1. MLP-Encoder** | Encoder for Synthetic/MoCap Data | |
| Input: $\mathbf{x}_{1:T}$ | Observed time series | BS $\times$ T $\times$ i_dim |
| Dense | 128 neurons, LeakyReLU | BS $\times$ T $\times$ 128 |
| Dense | 128 neurons, LeakyReLU | BS $\times$ T $\times$ 128 |
| Dense | 128 neurons, LeakyReLU | BS $\times$ T $\times$ 128 |
| Dense | Temporal embeddings | BS $\times$ T $\times$ z_dim |
| **2. MLP-Decoder** | Decoder for Synthetic/MoCap Data | |
| Input: $\hat{\mathbf{z}}_{1:T}$ | Sampled latent variables | BS $\times$ T $\times$ z_dim |
| Dense | 128 neurons, LeakyReLU | BS $\times$ T $\times$ 128 |
| Dense | 128 neurons, LeakyReLU | BS $\times$ T $\times$ 128 |
| Dense | i_dim neurons, reconstructed $\hat{\mathbf{x}}_{1:T}$ | BS $\times$ T $\times$ i_dim |
| **5. Inference Network** | Bidirectional Inference Network | |
| Input | Sequential embeddings | BS $\times$ T $\times$ z_dim |
| GRUInference | Bidirectional inference | BS $\times$ T $\times$ 2*z_dim |
| TemporalDynamics | Use past $\{\hat{\mathbf{z}}_{t-\tau}\}$ to infer posteriors of $\mathbf{z}_t$ | BS $\times$ T $\times$ 2*z_dim |
| ResidualBlock | Skip-connection of the two inferences | BS $\times$ T $\times$ 2*z_dim |
| Bottleneck | Compute mean and variance of posterior | $\mu_{1:T}, \sigma_{1:T}$ |
| Reparameterization | Sequential sampling | $\hat{\mathbf{z}}_{1:T}$ |
| **6. Causal Process Prior (VAR)** | Linear Transition Prior Network | |
| Input | Sampled latent variable sequence $\hat{\mathbf{z}}_{1:T}$ | BS $\times$ T $\times$ z_dim |
| InverseTransition | Compute estimated residuals $\hat{\epsilon}_{it}$ | BS $\times$ T $\times$ z_dim |
| SplineFlow | Score the likelihood of residuals | BS |
| **7. Causal Process Prior (NP)** | Nonlinear Transition Prior Network | |
| Input | Sampled latent variable sequence $\hat{\mathbf{z}}_{1:T}$ | BS $\times$ T $\times$ z_dim |
| InverseTransition | Compute estimated residuals $\hat{\epsilon}_{it}$ | BS $\times$ T $\times$ z_dim |
| JacobianCompute | Compute $\log\left(|\det\left(\mathbf{J}\right)|\right)$ | BS |
| SplineFlow | Score the likelihood of residuals | BS |

**Model Ablations** We start with BetaVAE and add our proposed modules successively as model variants. When the causal process prior network is added to the baseline, this model variant is equipped with the inference network and learned inverse transition functions. However, during the estimation of KL divergence in the causal process prior network, we use Mean Squared Error (MSE) directly, which corresponds to stationary Gaussian noise distribution, to replace the flow density estimators that evaluate the prior likelihood scores across nonstationary regimes. This variant shows the contributions of the learned causal transition functions. Furthermore, when the nonstationary flow estimator is added to the variant, this variant is implemented by removing the noise discriminator from LEAP while not changing any training settings.

Table C.2: Architecture details on CNN encoder and decoder. BS: batch size, T: length of time series, h_dim: hidden dimension, z_dim: latent dimension, F: number of filters, (Leaky)ReLU: (Leaky) Rectified Linear Unit.

| Configuration | Description | Output |
|---|---|---|
| **3.1.1 CNN-Encoder** | Feature Extractor | |
| Input: $\mathbf{x}_{1:T}$ | RGB video frames | BS $\times$ T $\times$ 3 $\times$ 64 $\times$ 64 |
| Conv2D | F: 16, BatchNorm2D, LeakyReLU | BS $\times$ T $\times$ 16 $\times$ 64 $\times$ 64 |
| Conv2D | F: 16, BatchNorm2D, LeakyReLU | BS $\times$ T $\times$ 16 $\times$ 64 $\times$ 64 |
| Conv2D | F: 32, BatchNorm2D, LeakyReLU | BS $\times$ T $\times$ 32 $\times$ 32 $\times$ 32 |
| Conv2D | F: 32, BatchNorm2D, LeakyReLU | BS $\times$ T $\times$ 32 $\times$ 32 $\times$ 32 |
| Conv2D | F: 64, BatchNorm2D, LeakyReLU | BS $\times$ T $\times$ 64 $\times$ 16 $\times$ 16 |
| Conv2D | F: 5 = number of objects | BS $\times$ T $\times$ 5 $\times$ 16 $\times$ 16 |
| **3.1.2 CNN-Encoder** | Keypoint Predictor | |
| Input: $\mathbf{x}_{1:T}$ | RGB video frames | BS $\times$ T $\times$ 3 $\times$ 64 $\times$ 64 |
| Conv2D | F: 16, BatchNorm2D, LeakyReLU | BS $\times$ T $\times$ 16 $\times$ 64 $\times$ 64 |
| Conv2D | F: 16, BatchNorm2D, LeakyReLU | BS $\times$ T $\times$ 16 $\times$ 64 $\times$ 64 |
| Conv2D | F: 32, BatchNorm2D, LeakyReLU | BS $\times$ T $\times$ 32 $\times$ 32 $\times$ 32 |
| Conv2D | F: 32, BatchNorm2D, LeakyReLU | BS $\times$ T $\times$ 32 $\times$ 32 $\times$ 32 |
| Conv2D | F: 64, BatchNorm2D, LeakyReLU | BS $\times$ T $\times$ 64 $\times$ 16 $\times$ 16 |
| Conv2D | F: 5 = number of objects | BS $\times$ T $\times$ 5 $\times$ 16 $\times$ 16 |
| Conv2D | SpatialSoftmax, lim=[-1,1,-1,1] | BS $\times$ T $\times$ 5 $\times$ 2 |
| **3.2 KiTTiMask-Encoder** | Mask Encoder | |
| Input: $\mathbf{x}_{1:T}$ | Semantic-segmented video frames | BS $\times$ T $\times$ 1 $\times$ 64 $\times$ 64 |
| Conv2D | F: 32, BatchNorm2D, ReLU | BS $\times$ T $\times$ 32 $\times$ 32 $\times$ 32 |
| Conv2D | F: 32, BatchNorm2D, ReLU | BS $\times$ T $\times$ 32 $\times$ 16 $\times$ 16 |
| Conv2D | F: 64, BatchNorm2D, ReLU | BS $\times$ T $\times$ 64 $\times$ 8 $\times$ 8 |
| Conv2D | F: 64, BatchNorm2D, ReLU | BS $\times$ T $\times$ 64 $\times$ 4 $\times$ 4 |
| Conv2D | F: h_dim, BatchNorm2D, ReLU | BS $\times$ T $\times$ h_dim $\times$ 1 $\times$ 1 |
| Dense | 2*z_dim neurons, features $\hat{\mathbf{x}}_{1:T}$ | BS $\times$ T $\times$ 2*z_dim |
| **4.1 CNN-Decoder** | Refiner | |
| Input: $\mathbf{z}_{1:T}$ | Sampled latent variable sequence | BS $\times$ T $\times$ 64 $\times$ 2 |
| ConvTranspose2D | F: 64, BatchNorm2D, LeakyReLU | BS $\times$ T $\times$ 64 $\times$ 32 $\times$ 32 |
| Conv2D | F: 64, BatchNorm2D, LeakyReLU | BS $\times$ T $\times$ 64 $\times$ 32 $\times$ 32 |
| ConvTranspose2D | F: 32, BatchNorm2D, LeakyReLU | BS $\times$ T $\times$ 32 $\times$ 64 $\times$ 64 |
| Conv2D | F: 32, BatchNorm2D, LeakyReLU | BS $\times$ T $\times$ 32 $\times$ 64 $\times$ 64 |
| Conv2D | F: 3, estimated scene $\hat{\mathbf{x}}_{1:T}$ | BS $\times$ T $\times$ 3 $\times$ 64 $\times$ 64 |
| **4.2 KiTTiMask-Decoder** | Mask Decoder | |
| Input: $\mathbf{z}_{1:T}$ | Sampled latent variable sequence | BS $\times$ T $\times$ z_dim |
| Dense | h_dim neurons | BS $\times$ T $\times$ h_dim $\times$ 1 $\times$ 1 |
| ConvTranspose2D | F: 64, BatchNorm2D, ReLU | BS $\times$ T $\times$ 64 $\times$ 4 $\times$ 4 |
| ConvTranspose2D | F: 64, BatchNorm2D, ReLU | BS $\times$ T $\times$ 64 $\times$ 8 $\times$ 8 |
| ConvTranspose2D | F: 32, BatchNorm2D, ReLU | BS $\times$ T $\times$ 32 $\times$ 16 $\times$ 16 |
| ConvTranspose2D | F: 32, BatchNorm2D, ReLU | BS $\times$ T $\times$ 32 $\times$ 32 $\times$ 32 |
| ConvTranspose2D | F: 1, estimated scene $\hat{\mathbf{x}}_{1:T}$ | BS $\times$ T $\times$ 1 $\times$ 64 $\times$ 64 |

## C.2 Hyperparameters and Training Details

We describe the hyperparameter selection criteria and discuss the impacts of hyperparameter values on the model performances. The training details are provided.

### C.2.1 Hyperparameter Selection and Sensitivity Analysis

The hyperparameters of LEAP include $[\beta, \gamma, \sigma]$, which are the weights of each term in the augmented ELBO objective, as well as the latent size $n$ and maximum time lag $L$. We use the ELBO loss on the validation dataset to select the best pair of $[\beta, \gamma, \sigma]$ because low ELBO loss always leads to high

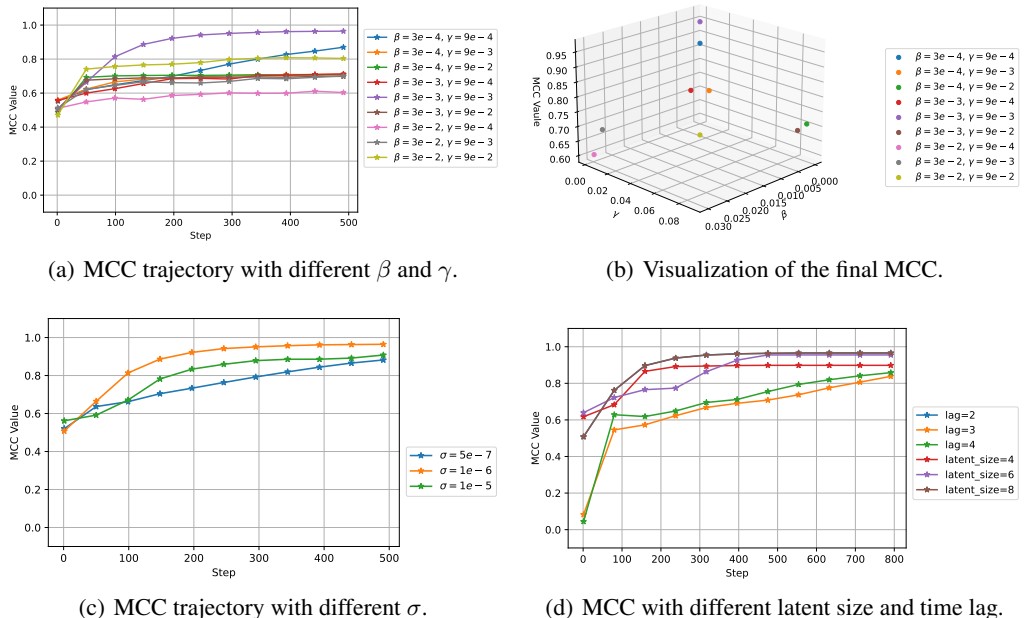

(a) MCC trajectory with different $\beta$ and $\gamma$.

(b) Visualization of the final MCC.

(c) MCC trajectory with different $\sigma$.

(d) MCC with different latent size and time lag.

Figure C.1: Impacts of hyperparameters on VAR dataset.

MCC. We always set a larger latent size than the true latent size. This is critical in video datasets because the image pixels contain more information than the annotated latent causal variables, and restricting the latent size will hurt the reconstruction performances. For the maximum time lag $L$, we set it by the rule of thumb. For instance, we use $L = 2$ for temporal datasets with a latent physics process.

However, it is known (Mita et al., 2021) that the performances of VAEs might change drastically as a function of the regularization strength. We thus conduct a sensitivity analysis on the impacts of hyperparameters on our identifiability performances. We report the MCC scores for the synthetic NP and VAR datasets on a hyperparameter grid. We found that the values of $\beta$ and $\gamma$ have a larger impact on the identifiability results, while the effects of $\sigma$ are relatively smaller. Furthermore, we have verified the robustness of our approach under different maximum time lags $L$ and latent size $n$ settings. The final MCC scores only show marginal differences, indicating that our approach is robust to the choices of $n$ and $L$. In summary, the performance of our approach is robust to the values of some of the hyperparameters, and for the remaining hyperparameters, we use separate validation data to set their values.

**Parametric (VAR) Dataset** We have performed a grid search of $\beta \in [3\text{E-}4, 3\text{E-}3, 3\text{E-}2]$ and $\gamma \in [9\text{E-}4, 9\text{E-}3, 9\text{E-}2]$ and reported the results in Fig. C.1(a). The best configuration is $[\beta, \gamma] = [3\text{E-}3, 9\text{E-}3]$. We plot the final MCC score as a function of the value of the two hyperparameters in Fig. C.1(b). For $\sigma$, we compare the MCC scores under different $\sigma \in [5\text{E-}7, 1\text{E-}6, 1\text{E-}5]$ with the optimal $(\beta, \gamma)$ value. The optimal configuration for $\sigma$ is 1E-6 as shown in Fig. C.1(c). Furthermore, we verify the robustness of our approach under different time lags $L \in [2, 3, 4]$ and latent dimensions $n \in [4, 6, 8]$ with $[\beta, \gamma, \sigma] = [3\text{E-}3, 9\text{E-}3, 1\text{E-}6]$ and the results are shown in Fig. C.1(d). We can see that the final MCC scores in all cases are around 0.9 with marginal differences, indicating that latent recovery performances of our approach is robust to the choices of $n$ and $L$.

**Nonparametric (NP) Dataset** Similarly, we have performed a grid search of $\beta \in [2\text{E-}4, 2\text{E-}3, 2\text{E-}2]$ and $\gamma \in [2\text{E-}3, 2\text{E-}2, 2\text{E-}1]$ on the NP dataset and reported the results in Fig. C.2(a). The best configuration is $[\beta, \gamma] = [2\text{E-}3, 2\text{E-}2]$. The final MCC scores under parameter grids of $[\beta, \gamma]$ are shown in Fig. C.2(b). The optimal configuration for $\sigma$ is 1E-6 in the search space $\sigma \in [1\text{E-}7, 1\text{E-}6, 1\text{E-}5]$ with the optimal $[\beta, \gamma]$ value, as shown in Fig. C.2(c). We verify the robust-

ness in terms of latent size $n \in [6, 7, 8]$ and the maximum time lags $L \in [1, 2, 3]$ with the optimal configuration $[\beta, \gamma, \sigma] = [\text{2E-3}, \text{2E-2}, \text{1E-6}]$ and the results are shown in Fig. C.2(d). The final MCC scores in all cases are around or higher than 0.85 with marginal differences.

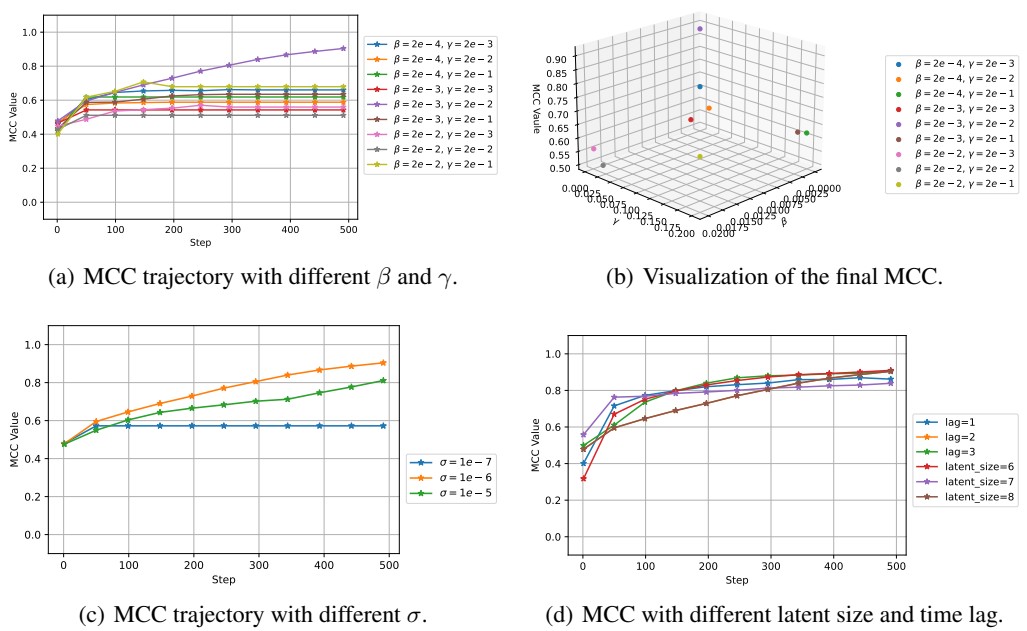

(a) MCC trajectory with different $\beta$ and $\gamma$.

(b) Visualization of the final MCC.

(c) MCC trajectory with different $\sigma$.

(d) MCC with different latent size and time lag.

Figure C.2: Impacts of hyperparameters on NP dataset.

### C.2.2 TRAINING

**Training Details**  The models were implemented in `PyTorch` 1.8.1. The VAE network is trained using AdamW optimizer for a maximum of 200 epochs and early stops if the validation ELBO loss does not decrease for five epochs. A learning rate of 0.002 and a mini-batch size of 32 are used. For the noise discriminator, we use SGD optimizer with a learning rate of 0.001. We have used four random seeds in each experiment and reported the mean performance with standard deviation averaged across random seeds.

**Computing Hardware**  We used a machine with the following CPU specifications: Intel(R) Core(TM) i7-7700K CPU @ 4.20GHz; 8 CPUs, four physical cores per CPU, a total of 32 logical CPU units. The machine has two GeForce GTX 1080 Ti GPUs with 11GB GPU memory.

**Training Stability**  We have used several standard tricks to improve training stability: (1) we use a slightly larger latent size than the true latent size for real-world datasets in order to make sure the meaningful latent variables are among the recovered latents; (2) we use AdamW optimizer as a regularizer to prevent training from being interrupted by overflow or underflow of variance terms of VAE; (3) we use a larger learning rate for the VAE than for the noise discriminator to prevent extreme extrapolation behavior of discriminator.

## D  ADDITIONAL EXPERIMENT RESULTS

### D.1  COMPARISONS BETWEEN LEAP AND BASELINES ON CMU-MOCAP DATASET

Because the true latent variables for CMU-Mocap are unknown, we visualize the latent traversals and the recovered skeletons, qualitatively comparing our nonparametric method with baselines in terms of the how intuitively sensible the recovered processes and skeletons are.

**Latent variable index**

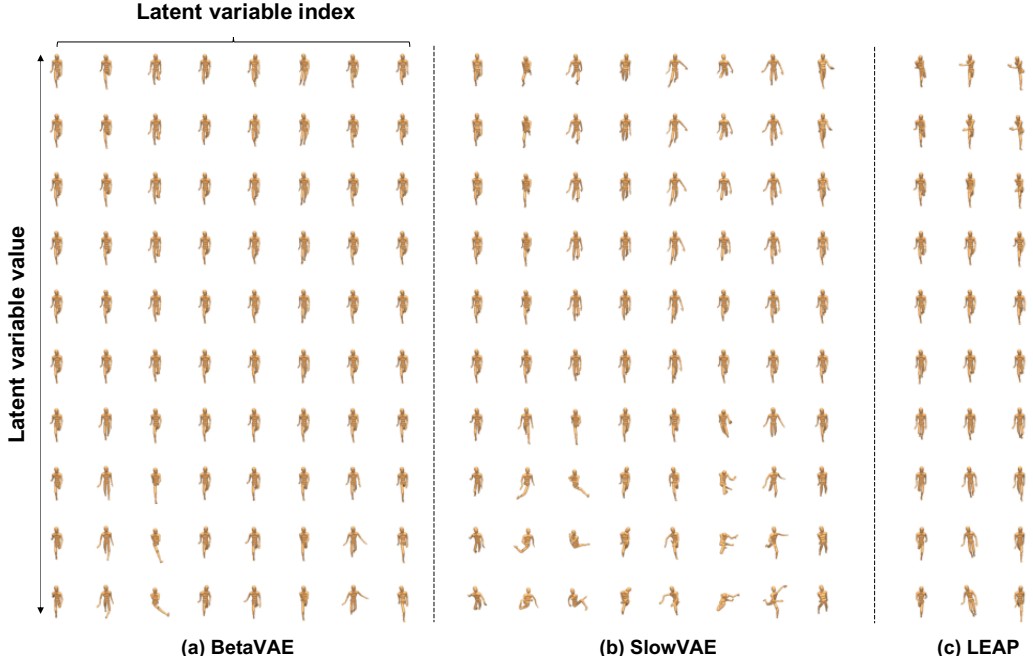

(a) BetaVAE       (b) SlowVAE       (c) LEAP

Figure D.1: Latent traversal comparisons between LEAP and the baselines. LEAP represents the data with causally-related factors, thus can represent the data with only much fewer latent variables (three vs eight) with smooth transitions dynamics. Video demonstrations are in: https://bit.ly/3kEVQhf.

**Latent Traversal** We fit LEAP and the baseline models using the same latent size $n = 8$ and the maximum time lags $L = 2$. As shown in Fig. D.1, LEAP represents the data with causally-related factors, thus explaining the data with fewer latent variables and smooth transitions dynamics. Only three latent variables are in fact used by LEAP and while the other five latent variables only encode random noise as seen from the video demonstration. BetaVAE and SlowVAE, however, need to use all the latent variables to represent the data. Furthermore, we find the three latent variables discovered by LEAP encode pitch, yaw, and roll rotations of walking cycles, which is close to how human beings perceive walking movement.

**Recovered Skeleton** As shown in Fig. D.2, LEAP recovers the cross relations between causal variables while BetaVAE and SlowVAE can only recover independent relations. The latent traversals of LEAP have shown that the three recovered latent variables may be the pitch, yaw, and roll rotations of the walk cycles. Therefore, the results of our approach indicate that the pitch (e.g., limb movement) and roll (e.g., shoulder movement) are causally-related while yaw has independent dynamics, which is closer to reality than the independent transitions discovered by BetaVAE and SlowVAE.

## D.2 MASS-SPRING SYSTEMS

We render the recovered latent variables using keypoint heatmaps in Fig. D.3(a). The learned representation successfully disentangles five objects in the scene, and the latent variables represent the horizontal and vertical locations of the balls. We further visualize the recovered skeletons from the estimated state transition matrices in Fig. D.3(b). The recovered skeleton is consistent with the underlying processes described in Fig. B.1(b) with SHD=0.

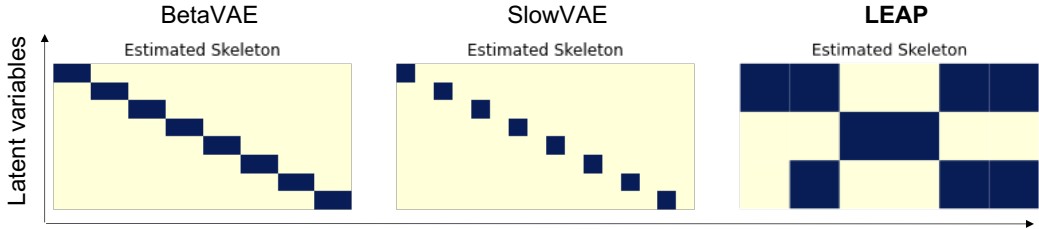

Figure D.2: Comparisons between LEAP and the baselines in terms of skeleton recovery. LEAP recovers cross relations between causal variables while baselines can only recover independent relations.

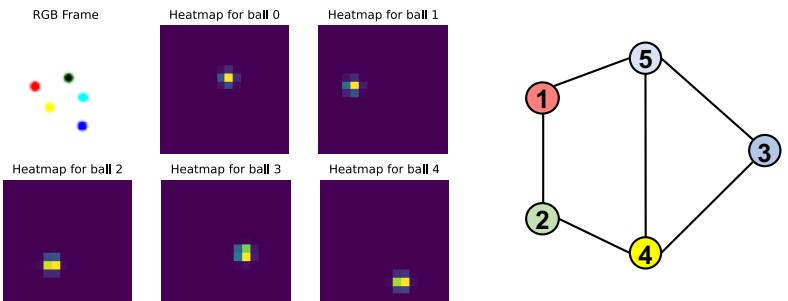

(a) Latent variables for a fixed video frame.  (b) Recovered causal skeletons.

Figure D.3: Visualization of recovered latent variables and the estimated skeletons for Mass-Spring system dataset.

# E EXTENDED RELATED WORK

Temporal dependencies and nonstationarities were recently used as side information $\mathbf{u}$ to achieve identifiability of nonlinear ICA on latent space $\mathbf{z}$. Hyvarinen & Morioka (2016) proposed time-contrastive learning (TCL) based on the independent sources assumption. It gave the very first identifiability results for a nonlinear mixing model with nonstationary data segmentation. Hyvarinen & Morioka (2017) developed a permutation-based contrastive (PCL) learning framework to separate independent sources using temporal dependencies. Their approach learns to discriminate between true time series and permuted time series, and the model is identifiable under the uniformly dependent assumption. Hälvä & Hyvarinen (2020) combined nonlinear ICA with a Hidden Markov Model (HMM) to automatically model nonstationarity without the need for manual data segmentation. Khemakhem et al. (2020) introduced VAEs to approximate the true joint distribution over observed and auxiliary nonstationary regimes. The conditional distribution in their work $p(\mathbf{z}|\mathbf{u})$ is assumed to be within exponential families to achieve identifiability on the latent space. A more recent study in causally-related nonlinear ICA was given by (Yang et al., 2021), which introduced a linear causal layer to transform independent exogenous factors into endogenous causal variables.

