# OpenReview forum: "Learning Temporally Causal Latent Processes from General Temporal Data"
_ICLR.cc/2022/Conference — ICLR 2022 Poster_

### Official Review · Reviewer_7eEa · 2021-10-18

**Correctness:** 3
**Technical Novelty And Significance:** 4
**Empirical Novelty And Significance:** 3
**Recommendation:** 8
**Confidence:** 3

**Main Review:**

Pros:
- In general, the topic of deriving rigorous conditions for the identifiability of temporal causal LVMs, and developing novel methods based on these conditions, is very interesting and relevant to the ML community.
-To my knowledge the setup considered in the article is novel (though see the cons below).
-The theoretical derivations seem correct (although I only glanced through the proofs in the supplement).
-The experiments are sufficiently thorough, including synthetic experiments and 3 real-world data sets.

Cons and questions to clarify:
- It seems that the (independence) assumptions, proofs, and the methodological ideas (e.g. contrastive learning to ensure independece) are rather close to previous works (e.g. by Klindt or by Hyvärinen and Morioka). The previous works have been properly cited in the text. Nonetheless, it would be beneficial for me to see a more concrete side-by-side comparison, to see where the novelty is located exactly (e.g. in the Supplement). For example, Eq. (2) and (4) very clearly present the current setup. Would it be possible to compare these with similar concise summaries of the previous setups?
-The method comes with many hyperparameters and it is not clear what their values are (e.g. beta, gamma, sigma in Eq. 11) or how they were selected (e.g. the latent dimension and the time lag in the experiments). Hence, judging how well the method really behaves in practice is difficult.
-Some assumptions are a bit abstract, for example the "Sufficient variability". How would a user of the method know if his/her data satisfy these assumptions? How do you know these assumptions are satisfied in your real world examples, and what happens if some of the assumptions are violated?
-The clarity of the presentation could be improved. For example, some figures are so small that it is impossible to make anything out of them when printed out (e.g. Fig. 8). Another concrete example: in the preliminaries of the proof for identifiability (section A.1) a sentence: "Since the learned unmixing function..." Why is g hat called the "unmixing function" here (and not elsewhere), what is g* and what is h (the sentence breaks in the middle before explaining h), and why the "we can assume..."? Similar small clarity and grammatical issues make the reading a bit cumbersome at places, although the technically the papers appears correct in general.

Very minor:
- Can you clarify where Eq. (6) comes from?
-In Fig 9, on the x-axis you have z_{t-1}^2 and z_{t-2}^2 twice. Is this a typo?


**Summary Of The Paper:**

The paper considers the learning of a temporal latent causal process: x_t=g(z_t), where x_t are observations, z_t are the underlying latent variables, and g is unknown. The paper makes two contributions: 1) presenting conditions under which the process can be identified (up to a permutation of the latent variables and their componentwise invertible transformations), and 2) developing a training framework that enforces the assumed conditions. The first contribution is theoretical and the second methodological. In addition, the paper presents empirical evidence for the ability of the method to recover the underlying causal process using synthetic experiments and real-world data.

Two cases are considered: 1) non-parametric, where a non-parametric transition is assumed in the latent space, and 2) parametric, where linear additive transitions are assumed. The main assumption for identifiability is the independent noise (IN) condition, which states that noise for the dimensions of z_t are independent of each other and across time steps t. This assumption is enforced through a contrastive approach during training. In addition, in the non-parametric case the noise is assumed non-stationary, such there exists sufficiently many auxiliary variables that affect the distribution of the non-stationary noise.


**Summary Of The Review:**

An interesting paper with relevant results on the identifiability of latent causal processes. I hope the authors could clarify my above concerns in their rebuttal.

POST-REBUTTAL UPDATE:
I found the authors had done a careful job addressing my original concerns. I don't have any remaining significant concerns. I find the article interesting and significant, and I support its acceptance at this stage. I will increase my score from 5 to 8.

---

> ### Author Response · Authors · 2021-11-22
> **Response to Reviewer 7eEa (Part I)**
>
> We appreciate that you read our paper very carefully and your informative feedback, which has helped improve our paper. Below please see our response to your concerns. In case you find anything unclear in the paper, please kindly let us know.
>
> Q1: It seems that the (independence) assumptions, proofs, and the methodological ideas (e.g. contrastive learning to ensure independence) are rather close to previous works (e.g. by Klindt or by Hyvärinen and Morioka). The previous works have been properly cited in the text. Nonetheless, it would be beneficial for me to see a more concrete side-by-side comparison, to see where the novelty is located exactly (e.g. in the Supplement). For example, Eq. (2) and (4) very clearly present the current setup. Would it be possible to compare these with similar concise summaries of the previous setups?
>
> A1: Thank you for this helpful suggestion. Certain modules of the proofs are shared by quite some investigations of the identifiability of underlying latent variables. However, please kindly note that our setting is inherently different from the two pieces of work you mentioned: PCL (Hyvärinen and Morioka) and SlowVAE (Klindt). PCL and SlowVAE assume the underlying sources are mutually independent for identifiability, and do not consider (time-delayed) causal relations between latent variables.
>
> To the best of our knowledge, this is one of the first works that successfully recover temporally causally-related latent processes from their nonlinear mixtures, while PCL and SlowVAE just recover independent processes. In light of your comment, we've included more thorough, side-by-side comparisons in Appendix A.4, by providing the mathematical formulations of their works and making comparisons in terms of problem setups and critical assumptions.  Please kindly let us know if you don’t find the explanations clear enough.
>
> References:
>
> [1] Hyvarinen, Aapo, and Hiroshi Morioka. "Nonlinear ICA of temporally dependent stationary sources." Artificial Intelligence and Statistics. PMLR, 2017.
>
> [2] Klindt, David, et al. "Towards nonlinear disentanglement in natural data with temporal sparse coding." arXiv preprint arXiv:2007.10930 (2020).
>
> Q2: The method comes with many hyper-parameters and it is not clear what their values are (e.g., beta, gamma, sigma in Eq. 11) or how they were selected (e.g., the latent dimension and the time lag in the experiments). Hence, judging how well the method really behaves in practice is difficult.
>
> A2: Thank you for pointing this out. We have included a new section in Appendix C.2.1 – Hyperparameter Selection and Sensitivity Analysis – in the revised draft to discuss hyperparameter selection criteria and the impacts of hyperparameters on the model performances. The summary of the results is quoted below.
> > “The hyperparameters of LEAP include $[\beta, \gamma, \sigma]$, which are the weights of each term in the augmented ELBO objective, as well as the latent size $n$ and maximum time lag $L$. We use the ELBO loss on the validation dataset to select the best pair of $[\beta, \gamma, \sigma]$ because low ELBO loss always leads to high MCC. We always set a larger latent size than the true latent size. This is critical in video datasets because the image pixels contain more information than the annotated latent causal variables and restricting the latent size will hurt the reconstruction performances. For the maximum time lag $L$, we set it by the rule of thumb. For instance, we use $L=2$ for temporal datasets with a latent physics process.”
>
> > “However, it is known (Mita et al., 2021) that the performances of VAEs might change drastically as a function of the regularization strength. We thus conduct a sensitivity analysis on the impacts of hyperparameters on our identifiability performances. We report the MCC scores for the synthetic NP and VAR datasets on a hyperparameter grid. We found that the values of $\beta$ and $\gamma$ have a larger impact on the identifiability results, while the effects of $\sigma$ are relatively smaller. Furthermore, we have verified the robustness of our approach under different maximum time lags $L$ and latent size $n$ settings. The final MCC scores only show marginal differences, indicating that our approach is robust to the choices of $n$ and $L$. In summary, the performance of our approach is robust to the values of some of the hyperparameters, and for the remaining hyperparameters we use the separate validation data to set their values.”
>
> References:
>
> [1] Mita, Graziano, Maurizio Filippone, and Pietro Michiardi. "An Identifiable Double VAE For Disentangled Representations." International Conference on Machine Learning. PMLR, 2021.

---

> ### Author Response · Authors · 2021-11-22
> **Response to Reviewer 7eEa (Part II)**
>
> Q3: Some assumptions are a bit abstract, for example the "Sufficient variability".
>
> A3: We are very grateful for the comment and suggestion. We've made the following efforts in the revised draft to address these issues. We've explained "Sufficient Variability" with precise definitions in the updated Appendix A.2.2 (It was not given in the original paper because it was rather long.) Intuitively, the condition says that the nonstationary regimes $\mathbf{u}$ must have a sufficiently complex and diverse effect on the transition distributions. In other words, if the underlying distributions are composed of relatively many domains of data, the condition generally holds true. For instance, in the linear Auto-Regressive (AR) model with Gaussian innovation where only the noise variance changes, the condition reduces to the statement in (Matsuoka et al., 1995) that the variance of each noise term fluctuates somewhat independently of each other across regimes.  A more complex example using modulated exponential family distribution is included in Appendix A.2.2. In case further explanations are helpful, please kindly let us know.
>
> References:
>
> [1] Kiyotoshi Matsuoka, Masahiro Ohoya, and Mitsuru Kawamoto. A neural net for blind separation of nonstationary signals. Neural networks, 8(3):411–419, 1995.
>
> Q4: How would a user of the method know if his/her data satisfy these assumptions? How do you know these assumptions are satisfied in your real-world examples, and what happens if some of the assumptions are violated?
>
> A4: Thanks for raising the concern. We've included the explanations of each assumption and discussed in Appendix A.2 how restrictive or mild they are in real applications.
>
> We've conducted more thorough ablation studies in Section 4.1 – robustness – to show the consequences of the violation of each of the assumptions to mimic real applications. How our approach is robust/sensitive to the assumptions is discussed. In summary, we find out that for (1) Vector Autoregressive (VAR) processes, our approach can gain partial identifiability with changing causal relations or low-rank state transitions but violating the IN condition and Generalized Laplacian Noise assumption distort the results, and for (2) Nonparametric (NP) processes, nonstationarity is necessary for identifiability. Furthermore, the differences of the MCC trajectories under 15 and 20 regimes seem marginal, suggesting that our approach does not always require at least $2n+1=17$ regimes to achieve full identifiability of the latent processes. The applicability of the assumed conditions has been further discussed in the updated Appendix A.2. For simulated data, we've considered the situation to violate each assumption, to cover more general cases, and produced detailed results. At the same time, on real-world datasets, we've tried to justify why the results produced by our approach are more convincing than those of the baselines.
>
> We hope these experimental results and explanations will resolve your concerns. Please kindly let us know if you don’t find them clear enough.
>
> References:
>
> [1] Kiyotoshi Matsuoka, Masahiro Ohoya, and Mitsuru Kawamoto. A neural net for blind separation of nonstationary signals. Neural networks, 8(3):411–419, 1995.
>
> Q5: The clarity of the presentation could be improved. For example, some figures are so small that it is impossible to make anything out of them when printed out (e.g., Fig. 8).
> A5: Thanks for your suggestions for improving the presentation. Specifically, in light of your comment, we have moved part of Fig. 8, which is not particularly relevant, to Appendix D.2, thus making the remaining parts more visible and informative.
>
> Q6: Another concrete example: in the preliminaries of the proof for identifiability (section A.1) a sentence: "Since the learned unmixing function..." Why is g hat called the "unmixing function" here (and not elsewhere), what is g*?
>
> A6: Thanks for pointing out the issues with notations. We realized that our previous usage of the notations should be improved, in light of your comment. Actually $g$ is the mixing function and this is a typo to say it is "unmixing function" (we apologize for the inconvenience caused.) Also, we use $g$ as the true mixing function and $\hat{g}$ as the estimated one.
>
> Q7: What is h (the sentence breaks in the middle before explaining h)
>
> A7: $h$ is the componentwise transformation that cannot be determined by the estimation procedures (it is what we call indeterminacy, and was defined in the first paragraph of the updated Appendix A.3.1.)

---

> > ### Comment · Reviewer_7eEa · 2021-11-29
> > **Thanks for the updates**
> >
> > Thanks for carefully considering my feedback. I updated my score to reflect my positive stance on this article (see the original summary).

---

> > > ### Author Response · Authors · 2021-11-29
> > > **Thanks for your feedback and updating the score**
> > >
> > > Dear Reviewer 7eEa,
> > >
> > > Thanks for your recognition of our efforts and kindly updating the score. We are very delighted to hear that all of your significant concerns have been resolved.
> > >
> > > With best regards,
> > >
> > > Authors of submission 481

---

> ### Author Response · Authors · 2021-11-22
> **Response to Reviewer 7eEa (Part III)**
>
> Q8: and why the "we can assume..."?
>
> A8: We've revised the equations in Appendix A.3.1 to illustrate how we can see $\hat{g}=g \circ h$ (we now use "see" instead of "assume"). The following statement was given in Appendix A.3.1:
> > “Since the learned mixing function $\mathbf{x}\_t = \hat{g}(\mathbf{z}\_t)$ can be written as $\mathbf{x}\_t = (g \circ (g)^{-1} \circ \hat{g}) (\mathbf{z}\_t)$ because of injective properties of $(g, \hat{g})$, we can see that $\hat{g} = g \circ \left((g)^{-1} \circ \hat{g}\right) = g \circ h$ for some function $h=(g)^{-1} \circ \hat{g}$ on the latent space.”
>
> Q9: Similar small clarity and grammatical issues make the reading a bit cumbersome at places, although the technically the papers appears correct in general.
>
> A9: We have carefully edited the paper multiple times in order to improve the clarity of the presentations. We hope you will find the revised paper pleasant to read. In case anything in the paper is unclear, please kindly let us know.
>
> Q10: Can you clarify where Eq. (6) comes from?
>
> A10: Eq. (6) directly uses ``change of variable formula’’ to derive the probability density function of the transformed variables. We've made where Eq. (6) comes from clear in the text above Eq. (6):
> > “By applying the change of variables formula to the map from $\mathbf{A}$ to $\mathbf{B}$ and because of the IN condition in Assumption 2 of Theorem 1, one can obtain the joint distribution of the latent causal variables as:”
>
> Q11: In Fig 9, on the x-axis you have $z\_{t-1}^2$ and $z\_{t-2}^2$ twice. Is this a typo?
>
> A11: Indeed this is a typo–thanks for spotting this! One of them should be $z\_{t-1}^3$ and $z\_{t-2}^3$. We've fixed the typo in Fig. 9 accordingly.

---

### Official Review · Reviewer_x73T · 2021-10-31

**Correctness:** 3
**Technical Novelty And Significance:** 3
**Empirical Novelty And Significance:** 3
**Recommendation:** 6
**Confidence:** 2

**Main Review:**

Please list both the strengths and weaknesses of the paper. When discussing weaknesses, please provide concrete, actionable feedback on the paper.

The specific strength of this paper is described above. Although technical contributions are derived from mixing the previous techniques, the problem setting seems to be completely new (but I am not an expert in this field), and experiments were sufficiently performed. Partly I have some specific comments as follows.

1. Sections 1 and 2: Parent (node) may be explicitly unclear. Is it the cause of the factor z?

2. Sections 1: follow follow

3. Definition 1: p_{\epsilon} is not defined in the introduction.

4. Fig 3c: It was unclear to me how to see Fig 3c. Is the negative correlation acceptable even if Figs 2a and 2b do not seem to have negative correlations or values (or ignore the signs?)

5. Fig 5: Some symbols were lost.

**Summary Of The Paper:**

The authors proposed a method to recover time-delayed latent causal variables and identify their relations from measured temporal data called Latent tEmporally cAusal Processes estimation (LEAP). This method leverages VAEs by enforcing our conditions through proper constraints in the causal process prior for recovering time-delayed latent processes from nonlinear mixtures without sparsity assumptions. The experimental results on synthetic datasets show that the proposed method outperformed baselines that do not leverage history or nonstationarity information. The results on public datasets including KiTTiMask, Mass-Spring System, and CMU MoCap demonstrate that temporally causal latent processes were reliably identiﬁed from observed variables under different dependency structures.


**Summary Of The Review:**

The problem setting seems to be completely new, experiments were sufficiently performed, the results show the superiority of the proposed method, whereas the technical contributions are derived from mixing the previous techniques.
Since I think the strengths in this paper totally outperformed the weaknesses, I provided a higher rating at this stage (but note that I am not an expert in this field).

---

> ### Author Response · Authors · 2021-11-22
> **Response to Reviewer x73T**
>
> We sincerely thank the reviewer for the time dedicated to reviewing this paper and the helpful comments. Below we give a point-by-point response to the comments.
>
> Q1: Sections 1 and 2: Parent (node) may be explicitly unclear. Is it the cause of the factor z?
>
> A1: Thank you for raising this question. Parent nodes $\mathbf{Pa}(z\_{it})$ denote the direct causes of the factor $z\_{it}$. We've given an explicit definition of "parent'' in Section 1 of the revised draft. A notation table is also included in Appendix A.1.
>
> Q2: Sections 1: follow follow
>
> A2: Thanks for pointing this out. We've fixed this typo in the revised draft.
>
> Q3: Definition 1: $p_{\epsilon}$ is not defined in the introduction.
>
> A3: Thanks for spotting this! $p\_{\epsilon}$ is the distribution of process noise $\epsilon$. We’ve given an explicit definition in the introduction and in the notation table of Appendix A.1.
>
> Q4: Fig 3c: It was unclear to me how to see Fig 3c. Is the negative correlation acceptable even if Figs 2a and 2b do not seem to have negative correlations or values (or ignore the signs?)
>
> A4: The “negative correlation” in Fig. 3c is due to the sign indeterminacy as part of the componentwise transformation that our approach cannot determine (it is what we call indeterminacy). To directly answer your question, this doesn’t conflict with the positive values or correlations in Fig 3a because Mean Correlation Coefficient (MCC) in Fig 3a computes the **absolute** value of the correlation coefficients. We've included a description of how MCC is computed in Appendix B.3. Similarly, the skeletons in Fig 3b indicate whether there is a causal edge (1) or not (0), thus taking only binary values. So Fig. 3b doesn’t conflict with Fig. 3c. (By the way, we believe you meant Fig 3a and 3b in the comment; please let us know if we are wrong.)
>
> Q5: Fig 5: Some symbols were lost.
>
> A5: Thank you for the feedback on the presentation. We've made Fig. 5 (and fonts) much larger than before and explained the symbols and colors in the legend. Please kindly let us know if you don’t find the revised figure clear enough.

---

> > ### Comment · Reviewer_x73T · 2021-11-27
> > **Thanks for the clarification**
> >
> > Thanks for the clarification.
> > I confirmed them and the modifications in the paper.

---

> > > ### Author Response · Authors · 2021-11-28
> > > **Thanks for the confirmation of clarity in the presentation**
> > >
> > > Thanks a lot for your encouraging feedback! We are very grateful that you went over our paper for a second time and confirmed the clarity in the presentation. Once again, thank you very much for the time dedicated to reviewing our paper!

---

> > > ### Author Response · Authors · 2021-11-29
> > > **Would you like to consider updating your recommendation?**
> > >
> > > Dear Reviewer x73T
> > >
> > > Thank you for your prompt feedback. Would you like to consider updating your recommendation, if your concerns are properly addressed?
> > >
> > > With best regards,
> > >
> > > Authors of submission 481

---

### Official Review · Reviewer_hJy9 · 2021-11-01

**Correctness:** 4
**Technical Novelty And Significance:** 3
**Empirical Novelty And Significance:** 2
**Recommendation:** 6
**Confidence:** 3

**Main Review:**

I think this paper is very promising and represents a step forward in the causal inference of temporal processes. However, there is a lack of clarity in the derivation of the model architecture that prevent future readers to use this paper at its full potential. I would be happy to increase my score if this issue is resolved.

### Model details

- The model derivation lacks clarity. I would not be able to reproduce the proposed architecture based only on the information given in the paper (appendix included). I think this is quite critical, even if the code is available online. In particular, Figure 2 has tildas on z, which are not to be found in the derivations of section 3 and this confusion seems still present in 3.1.1. I really encourage the authors to invest effort in making this architure more understandable, and especially highlight how each of the assumptions from theorems 1 and 2 are embodied.

- Again regarding clarity, the $\mathbf{u}$ are never defined nor properly introduced anywhere. This makes it harder to understand assumption 2 of Theorem 1. The IN condition, similarly, lacks a proper introduction (I assume it is the independent noise assumption).

- The sufficient variability condition is still a bit cryptic to me in terms of its implications. I think it would be nice to expand a bit on that as well.

### Experiments

- For the recovery of the causal relations, you compare the causal skeletons of the true and inferred process but it could you give a measure of the discrepancy between the causal graphs ? Such as minimum edit distance (up to a permutation of the true causal graph) ?

- The performance of the approach are very good compared to the baselines on the synthetic data. However, the data is generated as to fit the assumptions of Theorem 1 and 2. I would be good to also report the results of the baselines on the real-world datasets when the conditions are more general. Indeed, in this case, it's not clear if the proposed approach would as neatly outperform the baselines.

- In "Towards generalizing conditions" and in the conclusion, you seem to state that the only limitation of the method is that is assumes constant causal graph with no instantaneous relations. This is without counting on the assumptions laid out in the theorems. In particular, non-stationary noise and sufficient variablity. Of course, you need some conditions to be able to recover something but the impact of violating those conditions would be worth investigating.


**Summary Of The Paper:**

This papers proposes new conditions for identification of latent temporal causal processes, for both non parametric and parametric processes. They then propose an architecture emboyding the derived conditions such as to recover the latent temporal processes. The approach is shown to give better performance than compared baselines on synthetic data and to show good performance on non-synthetic datasets.

**Summary Of The Review:**

Very promising paper but there is some lack of clarity in the descriptions of the architure used, preventing future readers to use this paper at its full potential. I would be happy to increase my score if this issue is resolved. The baselines should also be run on the real-world datasets for comparison with the proposed approach when the data generating process does not exactly matches the assumptions made in this work.

---

> ### Author Response · Authors · 2021-11-22
> **Response to Reviewer hJy9 (Part I)**
>
> We are sincerely grateful to the reviewer for the informative feedback, which has helped improve our paper (please see the updated paper and appendix.) Please see our point-to-point response below.
>
> Q1: (Model details) The model derivation lacks clarity. I would not be able to reproduce the proposed architecture based only on the information given in the paper (appendix included). I think this is quite critical, even if the code is available online. In particular, Figure 2 has tildas on z, which are not to be found in the derivations of section 3 and this confusion seems still present in 3.1.1. I really encourage the authors to invest effort in making this architure more understandable, and especially highlight how each of the assumptions from theorems 1 and 2 are embodied.
>
> A1:
>
> 1. We apologize for the confusion. We realized that our previous usage of the notations should be improved, in light of your comment. We now use $\hat{\mathbf{z}}\_t$ to denote the estimated latent variables in Figure 2 and in the entire paper (no tilde is used).
>
> 2. The network architecture details are given in the updated Appendix C.1. The hyperparameter selection and sensitivity analysis are now discussed in the updated Appendix C.2.1. The training and implementation details are given in the updated Appendix C.2.2.
>
> 3. The connections between our architecture and our assumptions in the two conditions are highlighted in blue in Section 3.1.1 of the updated paper, where part (C) of the model is specified. We further explain the connections in the updated Appendix A.5 in the revised draft, which is quoted below. Please kindly let us know if you don’t find the explanations clear enough.
> > “As one can see from the proofs in Appendix A.3, what have been assumed for the estimation framework are the conditional factorial properties of $q(\hat{\mathbf{z}}\_t \vert \{\hat{\mathbf{z}}\_{t-\tau})\}, \mathbf{u})$ where $\hat{\mathbf{z}}\_t = h^{-1}(\mathbf{z}_t)$ and the model of temporal nonstationarities through nonstationary noises. The conditional factorial properties have been injected using the reparameterization trick in Eq. 7 with the IN condition in causal transition prior and the enforcing of spatiotemporal independence of estimated residuals through contrastive learning. The nonstationary noises are modeled with flow-based density estimators. We share the weights of the other modules (e.g., encoder, transition function, decoder, inference network, etc.) across nonstationary regimes while using separate flow models to estimate the density of residuals and evaluate the prior scores in each regime. We also use componentwise flow models so the learned residuals will not interact with each other in the estimation framework. Finally, in nonparametric processes, we warm-start the flow models to generate standard Gaussian noise. In parametric processes, the flow models are initialized to generate standard Laplacian noise. Note the other assumed conditions in the two theorems, such as Sufficient Variability and Nonsingular State Transitions, are data properties and do not need to be encoded as constraints in the estimation framework.”
>
> Q2: (Model details) Again regarding clarity, the u are never defined nor properly introduced anywhere. This makes it harder to understand assumption 2 of Theorem 1. The IN condition, similarly, lacks a proper introduction (I assume it is the independent noise assumption).
>
> A2: Thanks for raising this concern.  $\mathbf{u}$ denotes the nonstationary regime or domain index, as used in the recent work of nonlinear Independent Component Analysis (ICA), e.g., Khemakhem et al. (2020). We assume that we have observed the $|\mathbf{u}|$ regimes or domains of data, across which the noise distributions $p_{\epsilon_i \vert u}$ may change.
>
> We now explicitly define $\mathbf{u}$ at the very beginning (in Paragraph 2 of Introduction in the updated paper) and explain it again when we start to present our theoretical results (in Assumption 1 of Theorem 1). A detailed explanation of the Independent Noise (IN) condition is in Appendix A.2.1. In order to increase the readability of the proof, we've added a notation table in Appendix A.1. In case you don’t think it’s clear enough, please kindly let us know.
>
> References:
>
> [1] Khemakhem, Ilyes, et al. "Variational autoencoders and nonlinear ica: A unifying framework." International Conference on Artificial Intelligence and Statistics. PMLR, 2020.

---

> ### Author Response · Authors · 2021-11-22
> **Response to Reviewer hJy9 (Part II)**
>
> Q3: (Model details) The sufficient variability condition is still a bit cryptic to me in terms of its implications. I think it would be nice to expand a bit on that as well.
>
> A3: Thanks for raising the concern. We've explained "Sufficient Variability" with precise definitions in the updated Appendix A.2.2 (It was not given in the original paper because it was rather long.) Intuitively, the condition says that the nonstationary regimes $\mathbf{u}$ must have a sufficiently complex and diverse effect on the transition distributions. In other words, if the underlying distributions are composed of relatively many domains of data, the condition generally holds true. For instance, in the linear Auto-Regressive (AR) model with Gaussian innovation where only the noise variance changes, the condition reduces to the statement in (Matsuoka et al., 1995) that the variance of each noise term fluctuates somewhat independently of each other across regimes.  A more complex example using modulated exponential family distribution is included in Appendix A.2.2. In case further explanations are helpful, please kindly let us know.
>
> References:
>
> [1] Kiyotoshi Matsuoka, Masahiro Ohoya, and Mitsuru Kawamoto. A neural net for blind separation of nonstationary signals. Neural networks, 8(3):411–419, 1995.
>
> Q4: (Experiments) For the recovery of the causal relations, you compare the causal skeletons of the true and inferred process but it could you give a measure of the discrepancy between the causal graphs ? Such as minimum edit distance (up to a permutation of the true causal graph) ?
>
> A4: Thanks for this suggestion. In the revised draft, we have reported Structural Hamming Distance (SHD), which computes the number of edge insertions, deletions or flips in order to transform one graph to another on both synthetic and real-world datasets. SHD is one variant of Minimum Edit Distance in the causal discovery area by allowing only insertions, deletions and flips of edges. It was described in Section 4 – Evaluation Metrics – and explained in detail in Appendix B.3.
>
> References:
>
> [1] de Jongh M, Druzdzel M J. A comparison of structural distance measures for causal Bayesian network models[J]. Recent Advances in Intelligent Information Systems, Challenging Problems of Science, Computer Science series, 2009: 443-456.
>
> Q5: (Experiments) The performance of the approach are very good compared to the baselines on the synthetic data. However, the data is generated as to fit the assumptions of Theorem 1 and 2. I would be good to also report the results of the baselines on the real-world datasets when the conditions are more general. Indeed, in this case, it's not clear if the proposed approach would as neatly outperform the baselines.
>
> A5: Simulated data have ground-truth latent variables to verify whether latent variables are properly recovered and help conduct empirical comparisons. Most real-world datasets don't provide ground-truth latent variables. However, in the revised draft, we've shown the comparisons of our approach with baselines on three real-world datasets in Figure 6 and in Appendix D.1.
>
> 1. We first compare the MCC performances between our approach and baselines on KiTTiMask and Mass-Spring System dataset in Figure 6. Our approach considerably outperforms the other baselines that don't use history or nonstationarity information. On the KiTTiMask dataset, the gap between our result and that by SlowVAE is relatively small; this is because the latent processes on this dataset are rather independent and when the latent processes are independent, our parametric method reduces to SlowVAE, as its special case.
>
> 2. Because the true latent variables for CMU-Mocap are unknown, we visualize the latent traversals and the recovered skeletons in the updated Appendix D.1, qualitatively comparing our nonparametric method with baselines in terms of the how intuitively sensible the recovered processes and skeletons are. We also find that our approach can encode the data with fewer latent variables (3 vs. 8) with smooth transition dynamics, perhaps because our approach represents the data with causally-related factors. On the other hand, BetaVAE and SlowVAE can only recover independent processes. Video demonstrations of the comparisons are https://bit.ly/3kEVQhf.

---

> ### Author Response · Authors · 2021-11-22
> **Response to Reviewer hJy9 (Part III)**
>
> Q6: (Experiments) In "Towards generalizing conditions" and in the conclusion, you seem to state that the only limitation of the method is that is assumes constant causal graph with no instantaneous relations. This is without counting on the assumptions laid out in the theorems. In particular, non-stationary noise and sufficient variability. Of course, you need some conditions to be able to recover something but the impact of violating those conditions would be worth investigating.
>
> A6: We are very grateful to you for pointing this out. We've conducted more thorough ablation studies in Section 4.1 – robustness – to show the consequences of the violation of each of the assumptions. How our approach is robust/sensitive to the assumptions is discussed. In summary, we find out that for (1) Vector Autoregressive (VAR) processes, our approach can gain partial identifiability with changing causal relations or low-rank state transitions but violating the IN condition and Generalized Laplacian Noise assumption distort the results, and for (2) Nonparametric (NP) processes, nonstationarity is necessary for identifiability. Furthermore, the differences of the MCC trajectories under 15 and 20 regimes seem marginal, suggesting that our approach does not always require at least $2n+1=17$ regimes to achieve full identifiability of the latent processes. The applicability of the assumed conditions has been further discussed in the updated Appendix A.2. We hope these experimental results and explanations can resolve your concerns. Please let us know if they don’t properly address your concern.
>
> Q7: (Summary) Very promising paper but there is some lack of clarity in the descriptions of the architure used, preventing future readers to use this paper at its full potential. I would be happy to increase my score if this issue is resolved. The baselines should also be run on the real-world datasets for comparison with the proposed approach when the data generating process does not exactly matches the assumptions made in this work.
>
> A7: We appreciate your encouraging feedback. In the revised draft, we've (1) included clearer descriptions of the architecture in the main content, figures, and in Appendix C; (2) added comparisons with baselines on real-world datasets in Figure 6 of Section 4.2 and in Appendix D.1. For simulated data, we've considered the situation to violate each assumption, trying to mimic real situations, and produced detailed results. At the same time, on real-world datasets, we've tried to justify why the results produced by our approach are more convincing than those of the baselines (please refer to the response A5 above and Appendix D.1).

---

> ### Comment · Reviewer_hJy9 · 2021-11-27
> **Thank you for your work**
>
> Thanks you for taking my comments into account. In particular in the extension of the experiment setup and in improving the description of the method.

---

> > ### Author Response · Authors · 2021-11-28
> > **Thanks for increasing the rating score**
> >
> > Thanks a lot for checking our reply and our revised presentation. We have spent a lot of time addressing each reviewer's concerns. We are glad that you agree that our updated presentation and extended experiments have resolved your concerns regarding the clarity in the descriptions of the architecture, and the interpretability and robustness of the assumptions in the two model settings. Your valuable comments have improved our presentation and made our architecture more reproducible for the research community. Thank you very much!

---

### Official Review · Reviewer_Qhdx · 2021-11-12

**Correctness:** 4
**Technical Novelty And Significance:** 3
**Empirical Novelty And Significance:** 3
**Recommendation:** 6
**Confidence:** 3

**Main Review:**

I have some concerns:
1. The proof of the theorems are not clear. E.g., what is the condition u in Eq. (2) and the following many equations?
2. about the network structure: input is x_0, x_1, ...., x_T, and the latent is z_0, z_1, z_T, and the reconstruct is x_hat_0, ..., x_hat_T. The "time-delayed" is not clear in this structure. If there is time delay, how the x_0 can be reconstructed, and how is the reconstruction loss is calculated?
3. In Fig. 2, it is not clear what's the importance of each part of the model. How does the "C" part connect to the proposed two conditions? As the model seems big, what about the training stability?
4. The hyperparameters of the loss function are not clear. How to select the hyperparameters and their importance. The importance of each term of the loss is not very clear.
5. The relation of using masked input and the "sparse" is not clear. The mask is for sparse latent process, while it is also stated in the paper the method does not rely on sparsity in latent processes.
6. The structure of using "contrastive learning" and "flow" to ensure the identifiability are already exist in previous work by Khemakhem et al. I am not sure of the comparison with the previous work and the exact novelty.

**Summary Of The Paper:**

This paper propose two new conditions for nonlinear identifiability of temporal processes. The idea is very promising: discovery of conditional independent sources with time-delayed causal relations in between.
The proof and the code seem good.

**Summary Of The Review:**

The idea is very promising but some technical details are not very clear. Besides the proofs in the appendix, it is recommended to describe more about the architecture and implementation details in the appendix too, if length limited. It'e better also to add some discussion about how the conditions are satisfied in real applications and what to do if the conditions are not satisfied

---

> ### Author Response · Authors · 2021-11-22
> **Response to Reviewer Qhdx (Part I)**
>
> We greatly appreciate your thorough and constructive comments, many of which have helped improve our paper. Both the paper and appendix have been updated extensively to include as much detail as possible. Please see our point-to-point response to your concerns below.
>
> Q1: The proof of the theorems are not clear. E.g., what is the condition u in Eq. (2) and the following many equations?
>
> A1: We are very grateful to you for pointing this out.  $\mathbf{u}$ denotes the nonstationary regime or domain index, as used in the recent work of nonlinear Independent Component Analysis (ICA), e.g., Khemakhem et al. (2020). We assume that we have observed the $|\mathbf{u}|$ regimes or domains of data, across which the noise distributions $p_{\epsilon_i \vert u}$ may change. We now explicitly define $\mathbf{u}$ at the very beginning (in Paragraph 2 of Introduction in the updated paper) and explain it again when we start to present our theoretical results (in Assumption 1 of Theorem 1). A detailed explanation of our nonstationary regimes is in Appendix A.2.2. In order to increase the readability of the proof, we've added a notation table in Appendix A.1. In case you don’t think it’s clear enough, please kindly let us know.
>
> References:
>
> [1] Khemakhem, Ilyes, et al. "Variational autoencoders and nonlinear ica: A unifying framework." International Conference on Artificial Intelligence and Statistics. PMLR, 2020.
>
> Q2: About the network structure: input is $x_0, x_1, ...., x_T$, and the latent is $z_0, z_1, ..., z_T$, and the reconstruct is $\hat x_0, ..., \hat x_T$. The "time-delayed" is not clear in this structure. If there is time delay, how the $x_0$ can be reconstructed, and how is the reconstruction loss is calculated?
>
> A2: We have provided more details on the network structure and priors over temporally-earliest latents (e.g., $x_0$) in Paragraph (5) –Inference Network– in Appendix C.1. We have updated the structure plot (Fig. 2C) to highlight how the "time-delayed" information interacts with the model. In Fig. 2C,  the horizontal axis denotes the time index and the vertical axis denotes the variable element (channel) index. The time-delayed information is highlighted in the box on the left in Fig. 2C.
>
> To directly answer your question, when we consider lag 1 process, the time-delayed information of $z_{it}$ ($t \geq 1$) is $z_{i,t-1}$. For $z_{i0}$, there is no time-delayed information, and we use Gaussian $\mathcal{N}(0, 1)$ as its prior distribution.
>
> Please note that the mapping between $\mathbf{z}_t$ and $\mathbf{x}_t$ does not involve time delays. (Please note that bold $\mathbf{x}$ denotes the vector of all processes and the subscript denotes the time index, as defined in Paragraph 3 of Introduction.) Regarding the reconstruction loss, we use Mean Squared Error (MSE) for most of our datasets, except KiTTiMask, in which we use binary cross-entropy loss. We've included descriptions of reconstruction loss for each dataset in Section 3.3.

---

> ### Author Response · Authors · 2021-11-22
> **Response to Reviewer Qhdx (Part II)**
>
> Q3: In Fig. 2, it is not clear what's the importance of each part of the model. How does the "C" part connect to the proposed two conditions? As the model seems big, what about the training stability?
>
> A3: Thanks for kindly raising this concern.
>
> 1. Our theoretical identifiability results (Theorems 1 and 2) show the identifiability under the respective assumptions, and the assumptions are essential in the proof. Furthermore, in order to illustrate the importance of each part of the model empirically, we have provided the results of ablation studies in Table 2 of Section 4.1 of the submission and discussed the contributions of each part of the model to the final identification performances.
>
> In addition, in order to make the roles of each part of the model more explicit, we have updated the caption of Fig. 2 in the revised draft:
> > “Here  $\mathbf{x}\_{1:T}$ and $\mathbf{\hat{x}}\_{1:T}$ are the observed and reconstructed time series. LEAP consists of Encoders (A) and Decoders (D) with MLP or CNN for specific data types; (B) a bidirectional recurrent inference network that approximates the posteriors of latent variables $\mathbf{\hat{z}}\_{1:T}$, and (C) a causal process network that (1) models nonstationary latent causal processes $\mathbf{\hat{z}}\_t$ with Independent Noise constraint (see Theorem 1) or (2) models the linear transition matrix with Laplacian constraints (see Theorem 2).”
>
> 2. The connections between part (C) and the two conditions are highlighted in blue in Section 3.1.1 where part (C) of the model is specified. We further explain the connections in Appendix A.5 in the revised draft, which is quoted below.
> > “As one can see from the proofs in Appendix A.3, what have been assumed for the estimation framework are the conditional factorial properties of $q(\hat{\mathbf{z}}\_t \vert \{\hat{\mathbf{z}}\_{t-\tau})\}, \mathbf{u})$ where $\hat{\mathbf{z}}\_t = h^{-1}(\mathbf{z}_t)$ and the model of temporal nonstationarities through nonstationary noises. The conditional factorial properties have been injected using the reparameterization trick in Eq. 7 with the IN condition in causal transition prior and the enforcing of spatiotemporal independence of estimated residuals through contrastive learning. The nonstationary noises are modeled with flow-based density estimators. We share the weights of the other modules (e.g., encoder, transition function, decoder, inference network, etc.) across nonstationary regimes while using separate flow models to estimate the density of residuals and evaluate the prior scores in each regime. We also use componentwise flow models so the learned residuals will not interact with each other in the estimation framework. Finally, in nonparametric processes, we warm-start the flow models to generate standard Gaussian noise. In parametric processes, the flow models are initialized to generate standard Laplacian noise. Note the other assumed conditions in the two theorems, such as Sufficient Variability and Nonsingular State Transitions, are data properties and do not need to be encoded as constraints in the estimation framework.”
>
> 3. With our model, we have used the properties of the data in different aspects to constrain the estimation results and the learning process is rather stable in the sense that in most replications with random initializations, it converges to similar solutions. To illustrate that, we've used four random seeds in each experiment and reported the mean performance with standard deviation averaged across random seeds (e.g., in Fig. 3d). Furthermore, we've revised the draft to summarize the tricks we used for training stability in Appendix C.2.2 – Training stability. The revised text is quoted below.
> > “We have used several standard tricks to improve training stability: (1) we use a slightly larger latent size than the true latent size for real-world datasets in order to make sure the meaningful latent variables are among the recovered latents; (2) we use AdamW optimizer as a regularizer to prevent training from being interrupted by overflow or underflow of variance terms of VAE; (3) we use a larger learning rate for the VAE than for the noise discriminator to prevent extreme extrapolation behavior of discriminator.”

---

> ### Author Response · Authors · 2021-11-22
> **Response to Reviewer Qhdx (Part III)**
>
> Q4: The hyperparameters of the loss function are not clear. How to select the hyperparameters and their importance. The importance of each term of the loss is not very clear.
>
> A4: Thank you for pointing this out. We have included a new section in Appendix C.2.1 – Hyperparameter Selection and Sensitivity Analysis – in the revised draft to discuss hyperparameter selection criteria and the impacts of hyperparameters on the model performances. The summary of the results is quoted below.
> > “The hyperparameters of LEAP include $[\beta, \gamma, \sigma]$, which are the weights of each term in the augmented ELBO objective, as well as the latent size $n$ and maximum time lag $L$. We use the ELBO loss on the validation dataset to select the best pair of $[\beta, \gamma, \sigma]$ because low ELBO loss always leads to high MCC. We always set a larger latent size than the true latent size. This is critical in video datasets because the image pixels contain more information than the annotated latent causal variables and restricting the latent size will hurt the reconstruction performances. For the maximum time lag $L$, we set it by the rule of thumb. For instance, we use $L=2$ for temporal datasets with a latent physics process.”
>
> > “However, it is known (Mita et al., 2021) that the performances of VAEs might change drastically as a function of the regularization strength. We thus conduct a sensitivity analysis on the impacts of hyperparameters on our identifiability performances. We report the MCC scores for the synthetic NP and VAR datasets on a hyperparameter grid. We found that the values of $\beta$ and $\gamma$ have a larger impact on the identifiability results, while the effects of $\sigma$ are relatively smaller. Furthermore, we have verified the robustness of our approach under different maximum time lags $L$ and latent size $n$ settings. The final MCC scores only show marginal differences, indicating that our approach is robust to the choices of $n$ and $L$. In summary, the performance of our approach is robust to the values of some of the hyperparameters, and for the remaining hyperparameters we use the separate validation data to set their values.”
>
> References:
>
> [1] Mita, Graziano, Maurizio Filippone, and Pietro Michiardi. "An Identifiable Double VAE For Disentangled Representations." International Conference on Machine Learning. PMLR, 2021.
>
> Q5: The relation of using masked input and the "sparse" is not clear. The mask is for sparse latent process, while it is also stated in the paper the method does not rely on sparsity in latent processes.
>
> A5: Masked input is a regularization technique to prune/remove unnecessary input nodes to achieve sparsity of the causal relations, if causal relations are actually sparse. Kindly let us note that the sparsity regularization term is only used for interpretations of the causal relations but not for the estimation of the model, as stated in the first paragraph of Section 3.1.3. The statement meant that even if the underlying time-delayed relations are fully-connected, our approach can still identify the latent variables up to componentwise transformation. To validate this statement, we've experimented in ablation studies on a dataset with dense (fully-connected) causal relations (Table 2) and show that MCC can still achieve 0.983 on average. But if possible, one would like to use sparse relations for better interpretability by sparsity regularizations (e.g., with masked input).

---

> ### Author Response · Authors · 2021-11-22
> **Response to Reviewer Qhdx (Part IV)**
>
> Q6: The structure of using "contrastive learning" and "flow" to ensure the identifiability are already exist in previous work by Khemakhem et al. I am not sure of the comparison with the previous work and the exact novelty.
>
> A6: Thank you for asking this question. In light of your comment, we have included side-by-side comparisons in Appendix A.4 by providing the mathematical formulations of their work and made comparisons in terms of problem setups and critical assumptions. There are clear differences between our approach and iVAE (Khemakhem et al) from theoretical and methodological perspectives.
>
> First, in iVAE, the latent variables are conditionally independent, without causal relations in between. The supporting evidence is that iVAE failed on all the datasets in this paper which have (time-delayed) causally-related factors in the latent space (see, e.g., results in Fig. 3);
> Second, iVAE exploits the nonstationarity brought by side information (i.e., class label) on the prior distribution of latent variables $\mathbf{z}$. On the contrary, our nonparametric condition, instead of relying on the change in the prior distribution of latent variables, exploits the nonstationarity in the noise distribution, which is more natural in real-world datasets.
> Finally, iVAE assumes modulated exponential families while our nonparametric condition (Theorem 1) allows any kinds of modulation by side information $\mathbf{u}$ without those strong assumptions on the transition functions or distributions.
>
> In terms of architecture innovations, to remove distributional and functional form constraints in the nonparametric setting, we design a novel causal transition prior network by injecting the IN condition inside the reparameterization trick, which results in an efficient learnable transition prior mechanism that only requires computation of the determinant of a low-triangular Jacobian matrix. To the best of our knowledge, this structure has not been exploited in previous work yet.
>
> The closest work to our work might also include (1) PCL (Hyvarinen & Morioka, 2017) and (2) SlowVAE (Klindt et al., 2020). How our work can be differentiated from the three pieces of work (iVAE, PCL, SlowVAE) has been discussed in detail in Appendix A.4. To the best of our knowledge, this is one of the first works that successfully recover temporally causally-related latent processes from their nonlinear mixtures. Please kindly let us know if you don’t find the explanations clear enough.
>
> References:
>
> [1] Khemakhem, Ilyes, et al. "Variational autoencoders and nonlinear ica: A unifying framework." International Conference on Artificial Intelligence and Statistics. PMLR, 2020.
>
> [2] Hyvarinen, Aapo, and Hiroshi Morioka. "Nonlinear ICA of temporally dependent stationary sources." Artificial Intelligence and Statistics. PMLR, 2017.
>
> [3] Klindt, David, et al. "Towards nonlinear disentanglement in natural data with temporal sparse coding." arXiv preprint arXiv:2007.10930 (2020).
>
> Q7 (Summary): The idea is very promising but some technical details are not very clear. Besides the proofs in the appendix, it is recommended to describe more about the architecture and implementation details in the appendix too, if length limited. It'e better also to add some discussion about how the conditions are satisfied in real applications and what to do if the conditions are not satisfied.
>
> A7: Thanks for these suggestions. We have updated the manuscript thoroughly to make the technical details clearer; see, e.g.,
>
> 1. The network architecture details are included in Appendix C.1;
> 2. The hyperparameter selection and sensitivity analysis are discussed in Appendix C.2.1. The training and implementation details are given in Appendix C.2.2.
>
> We've included a new section in Appendix A.2 to explain and justify each critical assumption in the proposed conditions and discuss how restrictive or mild they are in real applications. In addition, we've conducted more thorough ablation studies in Section 4.1 – robustness – to show the consequences of the violation of each of the assumptions to mimic real applications. How our approach is robust/sensitive to the assumptions is discussed. In summary, we find out that for (1) Vector Autoregressive (VAR) processes, our approach can gain partial identifiability with changing causal relations or low-rank state transitions but violating the IN condition and Generalized Laplacian Noise assumption distort the results, and for (2) Nonparametric (NP) processes, nonstationarity is necessary for identifiability. Furthermore, the differences of the MCC trajectories under 15 and 20 regimes seem marginal, suggesting that our approach does not always require at least $2n+1=17$ regimes to achieve identifiability of the latent processes. Please also refer to the previous responses for specific points. We hope these explanations and additional experiments can resolve your concerns. In case anything is not clear, please kindly let us know.

---

> ### Author Response · Authors · 2021-11-29
> **Possible to provide your feedback soon so we can reply?**
>
> Dear Reviewer Qhdx,
>
> Thanks for your time and comments! Hope we are not bothering you, but we are looking forward to seeing whether our response and revision properly address your concerns and whether you have any further concerns, to which we hope for the opportunity to respond.
>
> We hope you will consider this work as an essential step towards unsupervised deep learning and disentanglement, especially in the scenario where temporal information is available.
>
> With best regards,
>
> Authors of submission 481

---

### Author Response · Authors · 2021-11-27
**[Last 3 days for us to respond] Could you please go through and comment on our response?**

Once again, thank you very much for the insightful comments. We have put a lot of efforts into providing the response and revising paper accordingly. Below let us summarize the new, informative experimental results inspired by your suggestions. As Nov 29 is the last day for us to respond, we hope for the chance to see and respond to your feedback. Thank you very much!

**Updated presentation**

* To reviewers **Qhdx** and **hJy9**: We have clarified the definitions of nonstationary regimes $\mathbf{u}$ at the very beginning (in Paragraph 2 of Introduction in the updated paper) and explained it again when we start to present our theoretical results (in Assumption 1 of Theorem 1).
* To reviewers **Qhdx** and **hJy9**: The network architecture details are given in the updated Appendix C.1. Hyperparameter selection and sensitivity analysis are now discussed in the updated Appendix C.2.1. The training and implementation details are given in the updated Appendix C.2.2
* To reviewers **Qhdx** and **hJy9**: The structure plot (Fig. 2) has been updated to highlight the “time-delayed” information. The connections between our architecture and our assumptions in the two conditions are highlighted in blue in Section 3.1.1 of the updated paper and further explained in the updated Appendix A.5 in the revised draft.

* To reviewers **Qhdx**, **hJy9** and **7eEa**: We have expanded the explanations of each assumption in the two model settings,  and discussed how restrictive or mild they are in real applications in Appendix A.2.

* To reviewers **Qhdx** and **7eEa**: We have included side-by-side comparisons in Appendix A.4 by providing the mathematical formulations of their work and made comparisons in terms of problem setups and critical assumptions.

* To reviewer **hJy9**: We have reported Structural Hamming Distance (SHD) as a measure of the discrepancy between the causal graphs. Descriptions of evaluation metrics (i.e., MCC and SHD) are in Appendix B.3.

* To reviewers **x73T** and **7eEa**: We have carefully edited the paper multiple times in order to improve the clarity of the presentations.

**Newly conducted experiments**

* To reviewer **hJy9**: We have provided the comparative results of our approach with baselines on three real-world datasets in Figure 6 and in Appendix D.
* To reviewers **Qhdx** and **7eEa**: We have included a new section in Appendix C.2.1 – Hyperparameter Selection and Sensitivity Analysis – in the revised draft to discuss hyperparameter selection criteria and the impacts of hyperparameters on the model performances.
* To reviewers **Qhdx**, **hJy9** and **7eEa**: We have conducted more thorough ablation studies in Section 4.1 – robustness – to show the consequences of the violation of each of the assumptions, to mimic real applications. How our approach is robust/sensitive to the assumptions is discussed.

---

### Public Comment · ~Weiran_Yao1 · 2022-03-13
**Code Release**

We thank all the reviewers for their constructive feedback! In order to promote reproducibility, we release our source code and models. We encourage the readers to run our experiments and update us on any encountered issues. The codebase is available at the following link: https://github.com/weirayao/leap.

Finally, we plan to merge this repo and our future research into the causal-learn package, and here (https://causal-learn.readthedocs.io/en/latest/) might be a good place to track our recent progress.

Thank you very much for your interest in our work!

Authors of submission 481

---

### Decision · Program_Chairs · 2022-01-20

**Decision:**

Accept (Poster)

**Comment:**

This paper proposes two new sets of conditions under which we can identify temporally causal latent processes. In this sense, this work makes valuable contributions to the theories of identifiability in this topic. The authors also propose LEAP, extending the VAE, to estimate temporally causal latent processes.

The reviewers had many constructive comments, and the authors strived to address them. In the end, the reviewers were satisfied with the final version of the paper.

Given that the theoretical identifiability theorems are major parts of the paper, I encourage the authors to elaborate more on the two sets of assumptions. They should discuss when these assumptions will hold and provide examples in which they will be violated.